# FROM MEMORIZATION TO REASONING IN THE SPECTRUM OF LOSS CURVATURE

## ABSTRACT

We characterize how memorization is represented in transformer models and show that it can be disentangled in the weights of both language models (LMs) and vision transformers (ViTs) using a decomposition based on the loss landscape curvature. This insight is based on prior theoretical and empirical work showing that the curvature for memorized training points is much sharper than non-memorized, meaning ordering weight components from high to low curvature can reveal a distinction without explicit labels. This motivates a weight editing procedure that suppresses far more recitation of untargeted memorized data more effectively than a recent unlearning method (BalancedSubnet), while maintaining lower perplexity. Since the basis of curvature has a natural interpretation for shared structure in model weights, we analyze the editing procedure extensively on its effect on downstream tasks in LMs, and find that fact retrieval and arithmetic are specifically and consistently negatively affected, even though open book fact retrieval and general logical reasoning is conserved. We posit these tasks rely heavily on specialized directions in weight space rather than general purpose mechanisms, regardless of whether those individual datapoints are memorized. We support this by showing a correspondence between task data's activation strength with low curvature components that we edit out, and the drop in task performance after the edit. Our work enhances the understanding of memorization in neural networks with practical applications towards removing it, and provides evidence for idiosyncratic, narrowly-used structures involved in solving tasks like math and fact retrieval.

## 1 INTRODUCTION

To what degree do models generate genuinely new knowledge, as opposed to simply reassembling snippets of data memorized from their training sets? Much discussion about the current utility and future prospects of large neural networks has centered on this question. On the one hand, a growing trophy case of model accomplishments on novel tasks near the frontier of human capabilities argues strongly against memorization strictly construed as an explanation of the full range of model behavior. But on the other hand, recent papers have argued convincingly that models do in fact memorize large volumes of their training data verbatim, and a surprisingly large fraction of naturalistic interactions with language-model chatbots contain significant verbatim recitations (Aerni et al., 2024; Carlini et al., 2022; Stoehr et al., 2024), behaviors which have significant implications for copyright and data privacy (Karamolegkou et al., 2023; Carlini et al., 2019; Shokri et al., 2017).

Models thus seem to have a significant and frequently used capacity for both memorization and generalization. The question is not whether models generalize or recite, but rather how these capabilities are represented, how they interact and trade off (Nguyen & Reddy, 2025), and how they might be modulated. These are the questions we seek to address in this work. We build on existing work that characterizes memorization in terms of the curvature of the loss landscape as a function of a model's weights (Foret et al., 2021; Hochreiter & Schmidhuber, 1997; LeCun et al., 1989; Hassibi et al., 1993; Keskar et al., 2017; Garg et al., 2024; Ravikumar et al., 2024; Jeon et al., 2024; Kim et al., 2023). This prior work argues theoretically and empirically that the loss landscape has highly curved directions in the neighborhood of memorized points, while generalization corresponds to flatter basins. We exploit this insight while extending it in several ways.

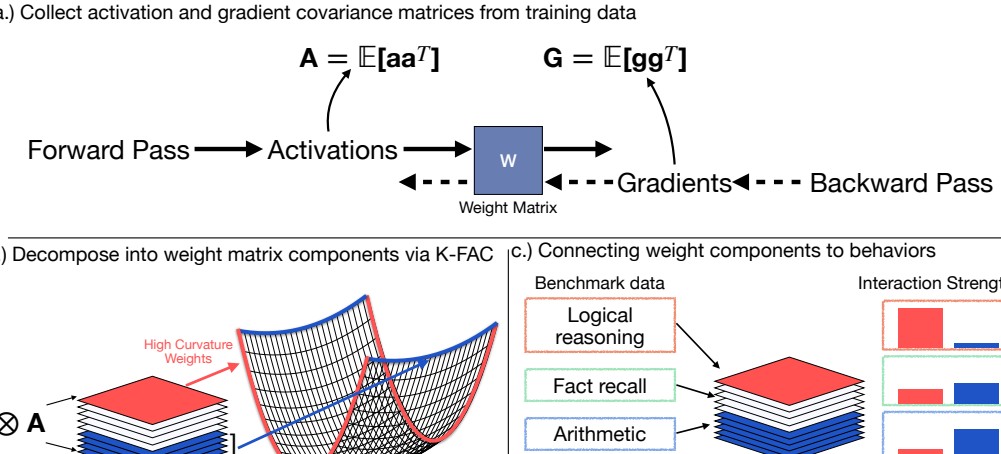

a.) Collect activation and gradient covariance matrices from training data

b.) Decompose into weight matrix components via K-FAC

c.) Connecting weight components to behaviors

Figure 1: Overview of our approach. We collect activations and gradients from a sample of training data (a), which allows us to approximate loss curvature w.r.t. a weight matrix using K-FAC (b). We decompose these weight matrices into components (each the same size as the matrix), ordered from high to low curvature. In language models, we show that data from different tasks interacts with parts of the spectrum of components differently (c).

**We study models' behavior in aggregate, rather than for individual examples.** Figure 1 (left). Are there model structures that account for memorization and generalization across large swaths of training data? We find that there are: in both language and vision models, the eigenbasis of the approximated Hessian of weight matrices uncovers distinct disentanglement of memorization and generalization, in a way that extends across a range of subdistributions of memorized data. In extending from studying per-example to bulk memorization, we propose a novel inversion of the previous interpretation of loss curvature: while individual memorized points are associated with high curvature, the direction of curvature varies across examples, meaning that, averaged across multiple examples, memorization directions are actually flatter than generalizing directions, which maintain a consistent moderate curvature across points.

**We propose an effective recitation-reduction technique based on ablating memorized directions in weight space.** We compare our results to a recent supervised memorization removal technique (BalancedSubnet (BSN); Sakarvadia et al. (2025)) and find that our method matches the suppression of the targeted forget set of BSN, and is similarly robust to stress tests (Huang et al., 2024), while removing far more unseen memorized data and achieving lower perplexity 5.2.

**We go beyond pure memorization and generalization and find curvature signatures of intermediate behaviors like fact retrieval and arithmetic.** Figure 1 (right). Many classical analyses of memorization and generalization have studied classification models, where the distinction between the two is clear. In modern language models, though, there is a much richer spectrum of behaviors between the poles of pure memorization (verbatim recitation of long passages) and pure generalization (de novo reasoning). For example, facts like "Paris is the capital of France" are memorized in the sense that they are specific pieces of information that the model knows, but are general in the sense that they are not tied to specific syntactic instantiations seen in the training data. Similarly, arithmetic and logical reasoning test a model's ability to generate novel inferences, but the inference rules and base axioms may be remembered from training. Going beyond prior work, we extend the loss curvature analysis to cover these reasoning types, situating them on a continuum between memorization and generalization. We find that, besides memorization, arithmetic and factual recall exhibit weight activation with low-curvature weight directions and are sensitive to their removal. On the other hand, logical reasoning, which does not necessarily require precise recall or calculation is robust to removing flat directions, which in some cases even improving performance. Our general approach is illustrated in Figure 1.

For all of our analyses, we measure curvature with the Kronecker-Factored Approximate Curvature (K-FAC) approximation to the model's Hessian, a computational technique that makes curvature analyses tractable at scale. K-FAC has been widely used for Hessian approximation in other settings, particularly for natural gradient optimization, but to our knowledge, our use of it to study memorization and generalization via curvature is novel.

## 2 RELATED WORK

Memorization, especially as a special case of overfitting, is a widely studied topic in both modern and classical machine learning. In the modern era of extremely large overparameterized models, there is particular interest in quantifying the ability and tendency of models to use their huge capacity to memorize training data. Recent work has shown that models are indeed able to store large amounts of data exactly (Morris et al., 2025), and that this data can be elicited verbatim in both naturalistic (Aerni et al., 2024) and adversarial (Carlini et al., 2022; Karamolegkou et al., 2023) regimes.

A closely related question is whether memories can be localized in model weights (Maini et al., 2023; Chang et al., 2024; Stoehr et al., 2024; Huang et al., 2024). Aligning with previous work (Hase et al., 2023; Karamolegkou et al., 2023), our work suggests that memorization is hard to pinpoint (and likely highly distributed), but we do find that distinctly loss-curved directions related to recitation of memorized data can be localized to some (early/late) layers. Similar localization work has studied the storage and retrieval of facts (Geva et al., 2021; Gur-Arieh et al., 2025; Meng et al., 2022; Dai et al., 2022; Rajamanoharan et al., 2023; Merullo et al., 2024; Menta et al., 2025), connecting to our analysis of factual recall in Section 6. Other work has focused on localizing functional components in weight space through other types of decompositions (Baker et al., 2025; Bushnaq et al., 2025).

Like the present study, previous work has used techniques like SVD to prune directions in weight space to compress models, expose low-rank structure, and understand memorization (Zhao et al., 2024; Jaiswal et al., 2025; Sharma et al., 2023). Relatedly, spectral dynamics has also been used explore memorization and generalization (Yunis et al., 2024).

Finally, a range of theoretical and empirical work has studied the connection between memorization and loss curvature, connecting high-curvature directions with memorized examples (Foret et al., 2021; Hochreiter & Schmidhuber, 1997; LeCun et al., 1989; Hassibi et al., 1993; Keskar et al., 2017; Garg et al., 2024; Ravikumar et al., 2024; Jeon et al., 2024; Kim et al., 2023). Bushnaq et al. (2024) investigate using the loss landscape for interpretability.

## 3 METHODS

### 3.1 FINDING MEMORIZATION WEIGHTS WITH K-FAC

In this work, we aim to decompose weight matrices in such a way that disentangles weight directions involved in verbatim memorization vs. generalization behavior. To do so, we decompose the MLP weight matrices in a model using the activations and gradients around them (Figure 1, top).

For a weight matrix $\mathbf{W}$, we collect a sample of activations going *into* $\mathbf{W}$ and backpropagate (using the loss over the model's distribution) to collect gradients on the output side of $\mathbf{W}$. We then form the covariance matrices $\mathbf{A}$ for activations and $\mathbf{G}$ for the gradients. Given an eignevector $\mathbf{u}$ from $\mathbf{G}$ and $\mathbf{v}$ from $\mathbf{A}$, the outer product $\mathbf{u} \otimes \mathbf{v}$ forms a rank-one matrix in the space of $\mathbf{W}$. We show that there is a strong ordered relationship between the eigenspectrum of these weight components, and memorized data; the relationship being that the components corresponding to the smallest eigenvalues are more likely to be used for reciting verbatim memorized training data.

The precise reason for *why* this construction makes sense to study memorization is because it corresponds to K-FAC (Kronecker-Factored Approximate Curvature; Martens & Grosse (2015)), which estimates Fisher Information Matrix (FIM) as $\mathbf{F} \approx \mathbf{G} \otimes \mathbf{A}$. Therefore, we refer to the weight components we use in this work $\mathbf{u} \otimes \mathbf{v}$ as **K-FAC** eigenvectors. Extensive prior work has connected loss curvature to memorization (§2), and K-FAC gives us a way to obtain essentially a dataset-average of curvature with respect to model weights. We detail this relationship in more detail in Section B.

## 3.2 COLLECTING K-FAC STATISTICS

We use K-FAC to approximate the Fisher block of each linear projection as a Kronecker product as shown in Equation 1, where $\mathbf{A}$ captures input correlations and $\mathbf{G}$ captures correlations of output gradients. We collect these factors for the MLP projections (gate_proj, up_proj, down_proj) by streaming $\sim$20M tokens from Dolmino/OLMo mixtures with sequence length 512 under next-token cross-entropy. In the forward pass we buffer pre-activation inputs $x$ (excluding the last position), and in the backward pass we record the corresponding gradients $g$. We accumulate $x^\top x$ and $g^\top g$ and normalize by the total number of contributing positions to form $\mathbf{A}$ and $\mathbf{G}$. For ViT experiments, we collect 10k images from the training split of ImageNet, using only the CLS token to collect activations and gradients.

## 3.3 MODELS

In this section, we describe the models we use for analysis, and settings we use when evaluating memorized data.

**Vision Transformers (ViTs)** Memorization in image classification models has been well studied, and there are simple recipes for producing models that memorize specific images. We train a family of 86M parameter ViT-Base models (Dosovitskiy et al., 2020) with 16x16 image patches at image resolution 224x224. We follow Dosovitskiy et al. (2020) training recipe on the ILSVRC 2012 ImageNet dataset (Russakovsky et al., 2015). In order to control memorization, we train ViT variants where a subset of training images have randomly assigned 'noised' labels. The only way for a model to reduce the loss on these images is to memorize these input-label pairs exactly. This is a standard setup for evaluating memorization in image classifiers (Zhang et al., 2017). Our default for evaluation is to train with 10% noised labels for 300 epochs. Our model trained with the noised labels achieves a top-1 accuracy on the validation set of 68.7%. When training with no noise, our model achieves 77.2% top-1 accuracy.

**Language Models (LMs)** We use the OLMo-2 family of models (OLMo et al., 2024), because they have openly accessible pretraining data and high performance on language modeling tasks. We report results for the 7B model. Previous work on evaluating memorization in LMs (Carlini et al., 2019; Huang et al., 2024; Shokri et al., 2017; Carlini et al., 2022) generally sampled sequences from a model's pretraining data, split each sequence into a prefix $P$ and suffix $S$, and evaluated whether the model produced $S$ under greedy decoding with prompt $P$. We adopt this same methodology, and use prefixes with length $|P| = 64$ tokens and suffixes with length $|S| = 48$ tokens.

## 4 DISENTANGLING WEIGHTS INVOLVED IN MEMORIZATION

This section will show that K-FAC is indeed a particularly good candidate to disentangle weights involved in memorized recitation. In simple terms, our procedure is to measure the interaction between memorized and non-memorized (clean) data with different K-FAC components of the weights. Our hypothesis is that if the curvature is a good measure of memorization, then the eigenvectors in different parts of the spectrum of the curvature basis (i.e., the top 10% of eigenvectors, bottom 50%, etc.) will activate differently from each other on memorized or clean data. The way we measure this is through activation ratios: for some hidden activation $x_{mem}$ stemming from a memorized input, we compare the ratio of its activation with a weight component $c$, which is some direction in the input space of the weight matrix to the activation with a clean input $x_{clean}$. If one weight component has a high $(c \cdot x_{mem})/(c \cdot x_{clean})$ ratio and another component has a very low ratio, then we know this weight matrix distinguishes the two types of data. To summarize the experiment: **for a given weight matrix, we would like to know if it has some components that interact more with memorized than non-memorized data**, with the hypothesis that the loss curvature basis is a principled way to disentangle these two signals.

**SVD** A reasonable idea is that we can disentangle memorization in the basis of singular vectors of a weight matrix, which decomposes into the components of most to least 'importance' for reconstructing the matrix. The top singular vectors may correspond to more general directions in weight space, and the lower ones towards directions used for reciting memorization (see Yunis et al. (2024)). If we

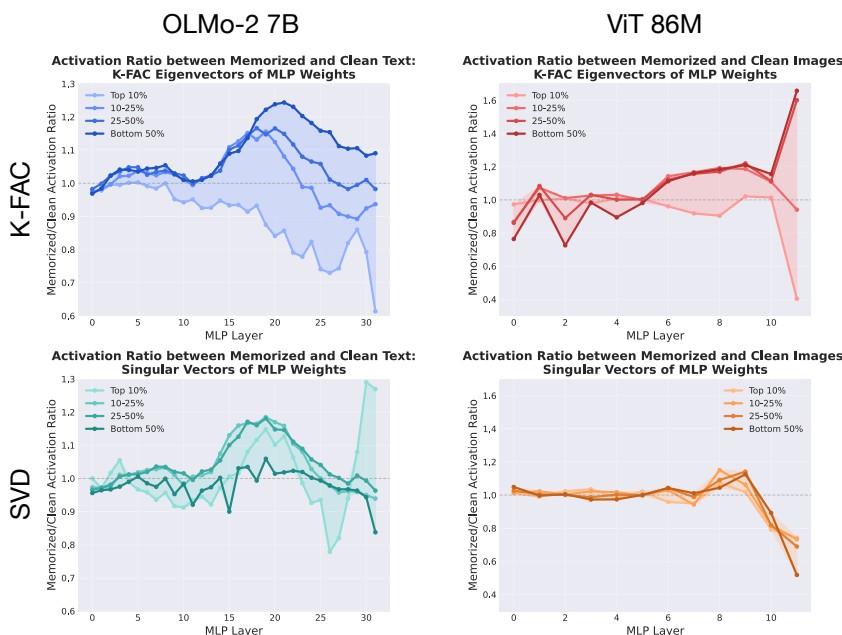

Figure 2: Large disentanglement in the weight space between memorized and non-memorized (clean) data, especially when decomposed into weight components with K-FAC (where the eigenspectrum also tends to sort by strength). The activation ratios show **selectivity**, where some parts of the spectrum activate more strongly for memorized data than others (or vice versa). The pattern is apparent in both LMs and ViTs.

compute the product between activations and weights as $\mathbf{W}x$, and the SVD is $\mathbf{USV^T}x$, then the right singular vectors in $\mathbf{V^T}x$ give us the magnitude with which those singular vectors read from.

**K-FAC Eigenvectors** Similar to the above setting with the right singular vectors, we can disentangle memorization in the basis of the activation eigenvectors. Since we do not measure gradient information in this setting, we project incoming activations onto the eigenbasis of $\mathbf{A}$. This can also be thought of as projecting onto the uncentered PCA components of activations, however, for the purposes of drawing percentile bands (top 10% vs. bottom 50%, e.g.) we do order the activation eigenvectors by the true FIM order[1].

**Results** Figure 2 shows our results. In both LMs and ViTs, K-FAC shows a more salient divergence between different parts of the eigenspectrum on memorized and clean data. For example, at the layer 22 MLP input in OLMo-7B, the bottom 50% of eigenvectors has a 23.1% higher activation on memorized data than clean data on average, and the top 10% of eigenvectors has a 26% higher activation on clean data than memorized data. Importantly, **the relative strength is sorted according to eigenvector band.** That is, in terms of activation with clean data, the strength of the top 10%>10-25%>25-50%>bottom 50%. SVD has a strong separation on the last layer (top 10% having 26% higher activation on memorized data), as well as around layer 20 in the gate projection only, but is lacking this sorted property. We therefore suggest that the curvature basis is more interpretable and accurate to describe the spectrum between memorized and non-memorized. In the ViT model we train, the top 10% eigenvectors have over a 2x activation strength on clean over memorized data at the last layer; the SVD for this model shows no such pattern. We train several other variants of the ViT models with varying weight decay, and find that it has a substantial effect on this separation with higher weight decay typically causing more drastic specialization. These results can be found in Appendix J.1.

These results support our hypothesis that the curvature basis allows us to disentangle weights involved in memorized recitation. As we will show in Section 6, we can use the amount of divergence between the top and bottom parts of the eigenspectrum to predict model behaviors on various tasks. In the

---

[1]That is, by the products of all combos of eigenvalues between activations and gradients.

following section, we will that removing weight components at the bottom of the K-FAC spectrum suppresses memorized data while retaining strong performance.

## 5 EDITING MODEL WEIGHTS TO SUPPRESS MEMORIZATION

A natural followup to finding distinct patterns of activations for (non-)memorized data across weight components in the curvature basis is whether we can use this discovery to prevent the recitation of memorized data while retaining general capabilities. We propose a novel editing method which projects an MLP weight matrix $\mathbf{W}$ into a subspace defined by its Hessian. As discussed in Section B, the eigendecomposition of this Hessian sorts components of $\mathbf{W}$ into directions of highest to lowest curvature in the loss landscape across a dataset. As we have discussed, in the aggregate across a dataset, the top eigenvectors correspond to generalizing directions; a claim that we will directly test here. Therefore, we propose keeping only the top $k\%$ of eigenvectors as a way to keep this shared structure while removing noisy or generally unimportant weight directions. In words, we define a matrix projection of an MLP weight matrix $\mathbf{W}$ that prevents communication through directions of low curvature in the eigenvectors of K-FAC.

Our method decomposes weight matrices using eigenbases derived from activation and gradient covariance matrices. Rather than truncating the eigenbases directly, we select specific pairs of eigenvectors whose joint contribution to curvature is highest, preserving a targeted fraction of the total curvature mass. Concretely, we start with K-FAC factor matrices $\mathbf{G} \in \mathbb{R}^{p \times p}$ (gradient covariance) and $\mathbf{A} \in \mathbb{R}^{q \times q}$ (activation covariance). These are decomposed into their eigenspaces as:

$$\mathbf{G} = \mathbf{U_G} \operatorname{diag}(\lambda) \mathbf{U_G}^\top, \quad \mathbf{A} = \mathbf{U_A} \operatorname{diag}(\mu) \mathbf{U_A}^\top, \quad \text{with} \quad \lambda_0 \geq \lambda_1 \geq \cdots \geq 0, \quad \mu_0 \geq \mu_1 \geq \cdots \geq 0.$$

Given a weight matrix $\mathbf{W} \in \mathbb{R}^{p \times q}$, we first express it in terms of these eigenbases as:

$$\mathbf{C} = \mathbf{U_G}^\top \mathbf{W} \mathbf{U_A}, \quad \text{where each coefficient} \quad C_{ij} = u_i^\top \mathbf{W} v_j.$$

To guide our compression, for each eigenvector pair $(i, j)$ we define a measure of curvature mass, $\Pi_{ij} := \lambda_i \mu_j$. The total curvature mass, summing over all pairs, is then given by:

$$M_{\text{tot}} := \sum_{i,j} \Pi_{ij} = \left( \sum_i \lambda_i \right) \left( \sum_j \mu_j \right).$$

Our compression strategy then selects a subset $S$ of eigenvector pairs, prioritizing those with the highest curvature mass. Formally, given a threshold parameter $\rho \in (0, 1]$, we construct $S$ by including pairs in descending order of $\Pi_{ij}$ until the cumulative curvature mass of selected pairs meets or exceeds the fraction $\rho$ of the total mass: $\sum_{(i,j) \in S} \Pi_{ij} \geq \rho M_{\text{tot}}$.

Once the subset $S$ is determined, we define a binary mask matrix $M \in \{0, 1\}^{p \times q}$, where $M_{ij} = 1$ if $(i, j) \in S$, and 0 otherwise. Finally, we construct the compressed weight matrix by zeroing out coefficients corresponding to pairs not in $S$:

$$\mathbf{W}_{\text{pairs}} = \mathbf{U_G} (\mathbf{C} \odot \mathbf{M}) \mathbf{U_A}^\top = \sum_{(i,j) \in S} C_{ij} u_i v_j^\top.$$

This method selectively preserves those directions in weight space most significant to the model's curvature.

We also test decomposing and truncating the bottom k% of singular values of a matrix $\mathbf{W}$, as well. It may be the case that the singular vector spectrum aligns incidentally with directions of high/low curvature, providing a data-free method for separating memorization. This setting also expands experiments on truncation explored in Yunis et al. (2024). Why might this alignment occur? Recall that we are approximating curvature using the eigenvectors of the covariance activations matrix $\mathbf{A}$ going into the layer. These eigenvectors are simply the uncentered principal component directions. when we say that the right singular vectors of $\mathbf{W}$ align with the top eigenvectors of $\mathbf{A}$, we're equivalently saying $\mathbf{W}$ places most of its sensitivity along the top input principal components. While we don't directly test this alignment, our results empirically support this interpretation.

## 5.1 EXPERIMENTAL SETUP

We construct two exact-match memorization sets under greedy decoding, one drawn from the pretraining corpus with a prefix and suffix of 48 and 64 respectively, and another of memorized historical quotes which we measure as memorized with a suffix of 8. Details are in Appendix C.

We report the metrics described in Section 5.1 separately on the Dolma and Quotes datasets. We primarily rely on strict accuracy to detect exact memorization in the dataset generation. However for evaluation, cases where strict = 0 but loose = 1 highlight sequences differing only slightly—semantically or syntactically—from the memorized target, representing partial memorization we also seek to avoid. Thus, loose accuracy complements strict accuracy by capturing near-verbatim memorization. Additionally, Levenshtein distance provides a continuous, threshold-free metric, allowing us to quantify memorization degradation more precisely.

**Metrics**   We evaluate memorization suppression in ViTs with three metrics: *memory reduction* (drop in top-1 predictions of memorized/noised labels), *ground-tuth recovery* (accuracy of recovering the true label for images trained with noised labels), and *validation accuracy* (post-edit validation accuracy to assess impact on core capabilities).

For LMs, given a prefix–suffix pair $(P, S)$ with $|S| = L$, we prompt with $P$ and greedily generate $L$ tokens to obtain $\hat{S}$. We compute the token-level Levenshtein distance $d(S, \hat{S})$ and report: *Strict Accuracy* $\mathbb{I}\big[d(S, \hat{S}) = 0\big]$; *Loose Accuracy* $\mathbb{I}\big[1 - d(S, \hat{S})/L \geq \tau\big]$ with $\tau = 0.75$; and *Average Normalized Distance* $\frac{1}{N} \sum_{n=1}^{N} \frac{d(S_n, \hat{S}_n)}{L_n}$, where higher values indicate less memorization.

**Baseline: Balanced Subnet (BSN)**   We compare to BSN, a recent memorization unlearning method introduced in Sakarvadia et al. (2025). This method trains a binary mask over individual MLP parameters optimized to maximize loss on a forget set (memorized data), while retaining low loss on a retain set (non-memorized, clean data).

**Model Settings**   For K-FAC: The full hyperparameter search details can be found in Appendix F. In LMs we edit layers 23, 24, and 25 at 60% energy retained in the up and gate projections in MLPs. In ViTs, we edit layers 0 and 11 to 75% energy on both up and down MLP projections.

For BSN: The best BSN settings for editing the language model were a loss weight of 0.7, 5 epochs, a sparsity ratio of 0.0015, and a learning rate of 0.3.

The best SVD settings for editing the language model were pruning ratios of 0.005 (0.5%) for the up and down projections and 0.5 (50%) for the gate projection in layer 21.

## 5.2 RESULTS

**K-FAC suppresses the broadest range of memorized text in LMs**   We compare our proposed K-FAC method against the state-of-the-art BSN baseline and SVD in Table 1. To ensure comparability of model coherence, we matched perplexities closely (K-FAC: 22.84, BSN: 23.59, SVD: 22.49), noting that BSN achieved slightly better nDCG@10 (0.97 vs. 0.91 for both K-FAC and SVD). While BSN required explicit training data, K-FAC and SVD did not—highlighting an important advantage of these approaches. On the Dolma validation set, K-FAC achieved 3.4% strict accuracy, BSN achieved 6.0%, and SVD achieved 3.0%. More notably, on the truly out-of-distribution historical quotes dataset, K-FAC achieved 16.1% strict accuracy, followed by SVD at 17.5% where as BSN achieved 60.0%. For completeness, Table 1 includes corresponding experiments on the 1B model, with settings detailed in Appendix H. In addition to perplexity, we include 20 generations from each method in Appendix L. Since it involves gradient ascent, BSN generates mostly nonsense when it detects memorization (and in some cases, for clean text). K-FAC and SVD edits retain very diverse generations; it is known, however, that low rank truncation, like that performed with the SVD edit can lead to unusual text, like dropped function words or incoherent text that don't show up in benchmark numbers (Sharma et al., 2023; JAISWAL et al., 2024). We only see 2 examples of this, but it may be necessary to train further after the edit to regain full expressivity. We don't see this issue with the K-FAC edits, which retain full rank. These results demonstrate our curvature-based pruning approach effectively mitigates memorization the best across both model sizes without requiring data to train a

| Method | Dolma Validation | | | Historical Quotes | | | Pile10k |
| | Strict (%) | Loose (%) | Avg Lev ↑ | Strict (%) | Loose (%) | Avg Lev ↑ | Perplexity ↓ |
|---|---|---|---|---|---|---|---|
| **7B Model** | | | | | | | |
| Baseline | 99.9 | 100.0 | 0.002 | 99.9 | 100.0 | 0.001 | 19.04 |
| BSN | 6.0 | 11.0 | 0.860 | 60.0 | 79.0 | 0.180 | 23.59 |
| K-FAC | 3.4 | 8.8 | 0.704 | 16.1 | 23.8 | 0.625 | 22.84 |
| SVD | 3.0 | 6.8 | 0.754 | 17.5 | 30.4 | 0.560 | 22.49 |
| **1B Model** | | | | | | | |
| Baseline | 98.46 | 99.38 | 0.005 | 98.5 | 98.95 | 0.006 | 23.19 |
| BSN | 3.0 | 5.0 | 0.900 | 57.0 | 66.0 | 0.250 | 25.41 |
| K-FAC | 2.8 | 7.2 | 0.761 | 27.7 | 39.9 | 0.470 | 26.53 |
| SVD | 3.2 | 6.3 | 0.781 | 39.6 | 48.5 | 0.401 | 26.94 |

Table 1: Comparison of unlearning methods on OLMo-2 7B and 1B models. Lower Strict/Loose percentages indicate better memorization suppression.

| Method | Memorized Train (%) ↓ | Train GT Accuracy (%) ↑ | Validation Accuracy (%) ↑ |
|---|---|---|---|
| Baseline | 81.6 | 10.5 | 67.0 |
| SVD | 3.5 | 58.9 | 67.8 |
| K-FAC | 3.5 | 66.5 | 71.7 |

Table 2: Comparison of edits on ViT-Base. K-FAC edits allow us to remove most memorization, recovers the ground truth label most of the time, and (likely through regularization) improves validation accuracy. The SVD (keeping only 5% of the selected K-FAC layers is also an effective baseline, but does a worse job recovering performance than K-FAC.

mask[2], achieving notably better generalization to unseen memorized content. Note that our sweep selects layers with very large gaps between memorized/clean activation ratio gaps across K-FAC eigenvectors; for example, we edit layers 23-25 in OLMo-2 7B, and layers 0 and 11 in ViT. These layers show very large gaps in Figure 2.

**ViTs Edited with K-FAC recover more ground truth labels than SVD**   Table 2 shows the results for editing ViT-Base with 10% training noise in various settings. On a per-layer basis, we see that pruning the earliest and latest layers provides the best results across the board. For both methods, we achieve the best performance when we prune MLPs 0 and 11 simultaneously, driving memorization performance down to 3.5% from over 80%. K-FAC also *increases* the validation accuracy over 4% from 67% to 71.7%, while SVD only increases performance around 1%. If we have successfully targeted memorized features, then we should see that the images that were memorized should switch to predicting their ground truth (GT) labels. K-FAC successfully raises the ground truth accuracy up to 66.5% while SVD reaches 58.9%.

**Stress tests**   Drawing from the positional perturbation stress tests outlined by Huang et al. (2024), we conducted a similar evaluation comparing K-FAC against BSN. For space we include these in Appendix I, but we find far less sensitivity to positional perturbations in both K-FAC and BSN than the older methods analyzed in prior work.

## 6   SPECTRUM OF MEMORIZATION TO REASONING IN DOWNSTREAM BEHAVIORS IN LMS

In traditional classification models, the distinction between memorization and generalization is stark and exhaustive: a label is either randomly generated (memorized) or inferred based on training (generalized). LMs have a varied landscape between memorization and reasoning. Here, we connect a wide range of LM behaviors to our story about weight space curvature and demonstrate tasks that have varying sensitivity to perturbations in weights. We demonstrate a spectrum of behaviors between pure memorization and pure reasoning that maps to our measurement of sharpness, and notably, that

---

[2]We use a sweep to find which layers to edit, which requires memorization labels to see if the edit is effective. With better understanding we may be able to pick which layers to edit without ever having labels for memorized sequences.

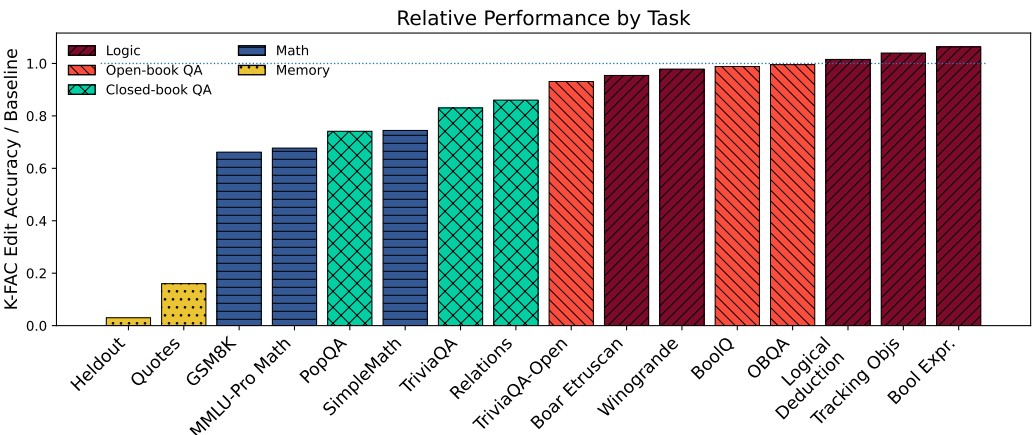

Figure 3: Sensitivity of different kinds of tasks to ablation of flatter eigenvectors. Parametric knowledge retrieval, arithmetic, and memorization are brittle, but openbook fact retrieval and logical reasoning is robust and maintain around 100% of original performance.

mathematical reasoning is highly brittle (Nikankin et al., 2025; Lindsey et al., 2025), while difficult non-numerical logical reasoning is among the most robust behaviors.

**Setup** We use the OLMES evaluation suite (Gu et al., 2025) to measure performance on edited models across benchmarks. We target benchmarks four main categories of tasks: **Closed-book fact retrieval**, **Open-book fact retrieval**, **Logical Reasoning**, and **Math** (arithmetic heavy). Open book retrieval involves question answering where there is a source available in context, whereas closed-book requires retrieval of facts directly from parametric knowledge. For example, TriviaQA is typically closed-book, but can be made open-book by including the relevant Wikipedia page (we refer to this as TriviaQA-Open). We include three non-standard datasets: Boar Etruscan (McCoy et al., 2023), which is an in-context constructed fake language like pig-latin. In order to perform well, models must rely solely on reasoning about rules provided in context, Relations (Hernandez et al., 2024), which is a dataset of factual relations such as "capital-of-country", and SimpleMath which tests two digit addition/subtraction problems (5-shot).

**Results** In Figure 3, we report a subset of benchmarks covering logic, fact recall, math, and our datasets of memorized sequences as a proportion of the unedited models accuracy. We find a mostly smooth drop off from logical reasoning (95-106% retention of baseline), open-book QA (93-99% retention), closed-book QA (74-86% retention), math (66-74%), and memorization (3-16%). Note that outside of the domains of math, and closed book QA, we find that K-FAC edited models perform very well compared to baseline (and often better than BSN), such as on CommonsenseQA, which doesn't cleanly fall into any of these categories. See Appendix K for details. The ranges of degradation we see reflect behavioral brittleness to weight perturbations specific to the domain of the task.

We can also show that this brittleness is measurable in terms of the magnitude of the activation along a K-FAC eigenvector direction (see §B). Figure 4 shows that interactions with the top and bottom of the curvature eigenbasis are predictive of how brittle they are (where a task falls relative to others in Figure 3). For example, hidden activations from OpenbookQA interact far less with the bottom of the eigenspectrum across layers 23-25 than the memorized data (memorized data interact at $\sim 1.6x$ higher magnitude); therefore, we might expect removing them to affect performance less, which is what we previously saw. The opposite is true for SimpleMath: hidden states interactions with the bottom part of the spectrum skew much more towards this dataset than clean data compared to the top of the spectrum, so we might expect removing them possibly harms performance. Again, this is consistent with the large drop seen in Figure 3). While we find that arithmetic ability is especially brittle, it's not clear from our results that LMs don't also contain delicate structure to solve it (Kantamneni & Tegmark, 2025). Additional discussion and results on math and factual recall are in Appendices E.1 and E.2. See further error analysis on fact retrieval and math in Appendix E.

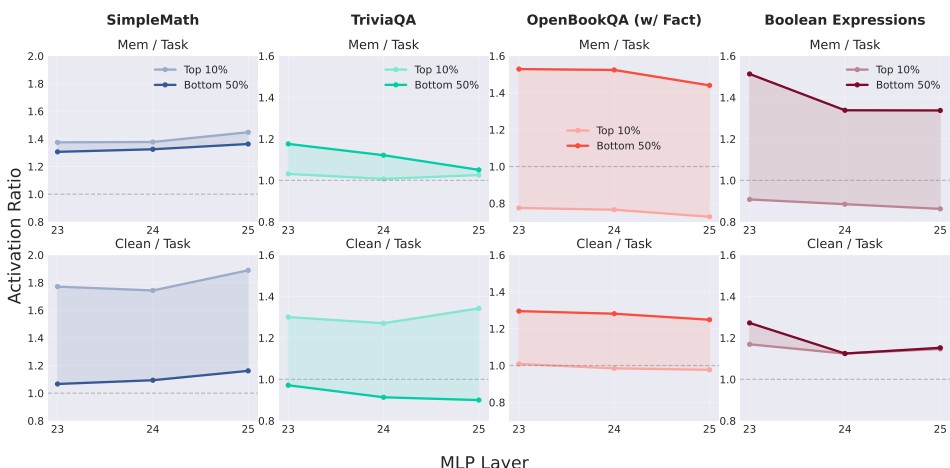

Figure 4: Eigenvector activation ratios for three different tasks compared against either clean or memorized data, visualized on layers we edit. The top 10% and bottom 50% bands of the curvature eigenbasis interact differently with memorized vs. non-memorized data. Large differences between these bands when comparing to memorized data (top row, openbookqa and bool. exprs.) indicate more resemblance to clean data processing, and large differences between these bands when comparing to clean data (bottom row, math and TriviaQA) indicates more similarity to memorization processing. The range of dissimilarity between each task and memorized/clean data matches very closely the behavioral degradation shown in Figure 3.

## 7 DISCUSSION AND LIMITATIONS

While we have made progress connecting behaviors to the loss curvature spectrum, our work has several limitations. We make no claims about fully removing memories from models, as our methods likely suffer from a tendency for memorized data to resurface after further tuning/perturbations (Lee et al., 2025). In terms of fully explaining why some tasks are more sensitive to perturbations and interact more with lower K-FAC eigenvectors, we have to speculate. For example, 'uncommon' weight directions which do not manifest as high curvature directions in K-FAC could correspond to precise and sophisticated structure (Kantamneni & Tegmark, 2025), rather than memorization or narrowly-useful patterns (Nikankin et al., 2025), necessarily. Our approximation of curvature is not perfect, and when estimating the bottommost eigenvalues, could suffer from numerical stability issues. While this doesn't affect our model edit, it may affect other analyses. We also do not consider cross-layer effect, which could be explored in future work.

## 8 CONCLUSION

We showed that loss-curvature provides a unifying lens for separating memorization from generalization in Transformers: the K-FAC curvature basis disentangles weight directions that support shared, reusable structure (top of the spectrum) from those that chiefly underwrite recitation and brittle behaviors (bottom of the spectrum). Leveraging this finding, we introduced a weight-editing method that preserves a targeted fraction of curvature mass and, across LMs and ViTs, strongly suppresses untargeted memorization while maintaining model coherence and, in the vision setting, even improving validation accuracy. Extending beyond verbatim recall, our analyses position downstream behaviors along a memorization–reasoning continuum: arithmetic and closed-book fact retrieval rely more on low-curvature directions and are disproportionately impacted by edits, whereas open-book and non-numerical logical reasoning are largely preserved or occasionally improved. These results (i) reconcile instance-level sharpness with population-level flatness, (ii) offer practical tools for recitation-reducing model editing, and (iii) go beyond previous results in finding curvature signatures for a range of model behaviors beyond strict memorization and generalization.

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

## A    Primer on the Eigendecomposition of A and G

This section provides background on how to think about the eigenvectors and eigenvalues of the Hessian, as approximated by the K-FAC factorization $\mathbf{F} \approx \mathbf{G} \otimes \mathbf{A}$. For a given weight matrix, recall that $\mathbf{A}$ is the covariance matrix of the activations going into it, and that $\mathbf{A} \in \mathbb{R}^{d_{\text{in}} \times d_{\text{in}}}$. $\mathbf{G}$ is the covariance matrix of the gradients on the output side of the matrix, and $\mathbf{G} \in \mathbb{R}^{d_{\text{out}} \times d_{\text{out}}}$.

Notice that we have $d_{\text{in}} * d_{\text{out}}$ eigenpairs in the Hessian. The approximate eigenvalues of the FIM are the products between each of the eigenvalues of the $\mathbf{G}$ and $\mathbf{A}$ matrices from K-FAC, and the corresponding eigenvectors are the Kronecker products between the eigenvectors of $\mathbf{G}$ and $\mathbf{A}$.

## B    Background on Loss Curvature and K-FAC

**Memorized individual instances exhibit sharp curvature**    Per-example analyses often find that memorized points are locally sharp, meaning the loss has a high second derivative in some directions (Ravikumar et al., 2024; Garg et al., 2024; Hochreiter & Schmidhuber, 1997; Foret et al., 2021). One way to think about this result is that the model is very brittle for that point: if a model memorized a datapoint exactly, and you were to perturb either the input itself, or weights interacting with it, the loss would spike (since the model can no longer recognize the *exact* point it memorized). Note that this is a simplified view, and that models rarely models using better generalizing mechanisms may be more robust to perturbations and the loss is locally flatter. We can quantify how curved the loss landscape is by measuring how *quickly* the sharpest direction of the Hessian is (through its top eigenvalue) or how *much* curvature is present in the Hessian (the trace). This way of measurement establishes that there are **directions**[3] of high and low curvature that we can use to detect memorization. The remainder of this section will cover the connection between K-FAC and a dataset-average picture of loss curvature, and discuss how this picture inverts this intuition about individual points.

**K-FAC's relationship to curvature**    Following Martens & Grosse (2015); Foret et al. (2021), we study memorization and generalization through the lens of loss curvature, specifically as a function of the model's weights (Keskar et al., 2017). Mathematically, the curvature of the loss landscape is captured by the Hessian $\mathbf{H} = \nabla_\theta^2 L(\theta)$, where $L$ is the loss function and $\theta$ is the vector of flattened model weights. Practically, though, $\mathbf{H}$ is not tractably computable for any but the smallest models, as its size is quadratic in the number of model weights. Prior work bypasses explicitly computing $\mathbf{H}$ by approximating its top eigenvalues and/or trace Ghorbani et al. (2019); Foret et al. (2021); Ravikumar et al. (2024); Garg et al. (2024)[4], or by making other approximations Hochreiter & Schmidhuber (1997); Keskar et al. (2017). For our analyses, though, we need a more complete picture of the whole spectrum of $\mathbf{H}$, and to get this picture, we turn to the Kronecker-Factored Approximate Curvature (K-FAC) Martens & Grosse (2015). Originally introduced as an efficient natural-gradient method for optimization, K-FAC approximates the Fisher Information Matrix (FIM) and provides a structured approximation to the loss curvature without forming the full Hessian.

---

[3]directions in whatever space you are deriving with respect to. In our case, weight space.

[4]Ravikumar et al. (2024); Garg et al. (2024) measure the trace w.r.t. the *inputs* rather than parameters, but this distinction isn't important for this point.

For a model trained with softmax cross-entropy loss, the relationship of the FIM $\mathbf{F}$ to the curvature of parameters is given by:

$$\mathbf{F} = \mathbb{E}_D[\nabla_\theta \log p_\theta(y \mid x, \theta) \nabla_\theta \log p_\theta(y \mid x, \theta)^T] = \mathbb{E}_D[\nabla_\theta^2(-\log p_\theta(y|x))]$$

Here, $D$ is a dataset consisting of input-label pairs $(x, y)$, and $p_\theta(y \mid x)$ is the model's predicted label distribution for input $x$. For an individual matrix $W \in \mathbb{R}^{d_{\text{out}} \times d_{\text{in}}}$ with incoming activations $a$ and backpropagated gradients $g$, K-FAC gives an easily computable approximation to a weight matrix $\mathbf{W}$'s block of $\mathbf{F}$:

$$\mathbf{F}_W \approx \mathbf{G} \otimes \mathbf{A} = \mathbb{E}[gg^T] \otimes \mathbb{E}[aa^T], \tag{1}$$

where $\mathbf{A} \in \mathbb{R}^{d_{\text{in}} \times d_{\text{in}}}$ and $\mathbf{G} \in \mathbb{R}^{d_{\text{out}} \times d_{\text{out}}}$. In words, this is the Kronecker product of the (uncentered) second-moment matrices of the activations going into the layer and the gradients coming out. When computing the loss to backpropagate into $\mathbf{G}$, we sample $\hat{y}$ from the model's predicted label distribution, rather than taking the ground truth $y$. Not only is this important for the correctness of the FIM FIM(Martens & Grosse, 2015), but it also means we can use this method without any labeled data.

**Instance- vs. population-level curvature.** Our use of the Fisher/K-FAC differs by averaging curvature across data, which emphasizes directions that are *consistently* important. Idiosyncratic sharp directions associated with specific examples point in different directions and largely cancel in the average, contributing to a low-curvature background. Directions that implement shared mechanisms (used by many inputs) add coherently and remain high-curvature on average. This explains why retaining high curvature mass preserves general abilities, while removing low-curvature components preferentially suppresses recitation.

**Decomposing K-FAC** Figure 1 (left). Throughout this work, we will decompose the FIM of a weight matrix $\mathbf{W}$ into distinct components (directions) to analyze their role in reciting memorized data. We can compute the eigendecomposition of the FIM by individually eigendecomposing $\mathbf{A}$ and $\mathbf{G}$ (see Appendix A). We refer to the eigendecomposition of K-FAC as the curvature basis, since the eigenvalues are sorted in terms of most to least loss curvature. **A single eigenvector of K-FAC is the outer product of an activations eigenvector and gradient eigenvector, and thus can be considered a weight component of $\mathbf{W}$ (i.e., as a matrix with $\mathbf{W}$'s size).** Therefore, when we describe the 'activation' of data with a rank-one component $\mathbf{C}$, we are describing the matrix vector product $\mathbf{C}x$. The magnitude of this product would be the norm of the resulting vector.

## C DATASETS

**Dolma** We mine memorized continuations from Dolma to obtain on-distribution memorization examples. Fixed-length windows $[64 \mid 48]$ are sampled per document and a window is labeled memorized iff the teacher-forced argmax at each of the $48$ suffix positions equals the gold token. Positives are aggressively deduplicated to avoid inflation from near-identical suffixes (e.g., templatic code/comments). The resulting 1000 sequences are split evenly: one half trains BSN (unlearning) and sweeps K-FAC, and the other half validates both.

**Historical Quotes** For each quote (length $\geq 9$ tokens), the prefix is all but the last $8$ tokens and the suffix is the final $8$. We greedily generate $8$ tokens from the prefix and mark memorized on exact match. As quotes have canonical phrasing and high surface regularity, an exact-match is meaningful and less sensitive to trivial paraphrases. This $512$ dataset is used strictly for validation to check whether methods preserve non-target knowledge while removing targeted memorization.

## D NDCG@10 (TOKEN-RANKING OVERLAP)

For each token position $t$, the frozen baseline provides a ranked list of its top-$K$ next-token predictions, $B_t = \{b_{t,1}, \ldots, b_{t,K}\}$. After editing, the model produces its own top-$K$ ranking $\hat{y}_{t,1:K}$. We assign graded relevance scores based on the presence and rank order of the edited model's predictions within the baseline set:

$$\text{rel}(r) = \begin{cases} K - r + 1, & \text{if } \hat{y}_{t,r} \in B_t, \\ 0, & \text{otherwise.} \end{cases}$$

We then compute the Discounted Cumulative Gain (DCG), normalized by the Ideal DCG (IDCG), resulting in the normalized Discounted Cumulative Gain (nDCG):

$$\text{DCG}_t = \sum_{r=1}^{K} \frac{\text{rel}(r)}{\log_2(r+1)}, \quad \text{IDCG}_K = \sum_{r=1}^{K} \frac{K-r+1}{\log_2(r+1)}, \quad \text{nDCG}_t = \frac{\text{DCG}_t}{\text{IDCG}_K} \in [0,1].$$

We compute nDCG@10 on the first 200k tokens of the held-out `pile10k` dataset. Intuitively, this measures how closely the edited model's token-ranking aligns with the baseline, capturing *local preference drift*. We specifically chose ranking rather than probabilities to isolate the ordering of high-probability tokens, as these largely determine predictive entropy. We report the mean nDCG@10 across positions (higher is better) as an indication of how faithfully the model preserves the baseline's preference structure post-edit.

# E ERROR ANALYSIS ON LM BENCHMARKS

## E.1 ERROR ANALYSIS ON MATH DATA

We find that arithmetic is specifically hurt by the K-FAC edit. This could be because arithmetic problems themselves are memorized (at the 7B scale), or because they require narrowly used directions to do precise calculations. We find it interesting, though, that this is so specifically and negatively affected when seemingly related skills remain intact. Besides the logical reasoning benchmarks included in Figure 3, we find that while MMLU-Pro Math drops to 67% of baseline (23.4% to 15.8%), the computer science subset stays at about the same level (Baseline: 26.1%, K-FAC: 26.3%, BSN: 25.1%). An example from GSM8K also shows that despite generating the *exact* same chain of thought as the baseline model, it failed specifically at making the calculation (Figure 3. Not all generations are as similar to the original, but this demonstrates how targeted the new deficit for arithmetic is. We include an example of essentially the opposite (same generation but fixing an error) in Boolean Expressions; see Table 4.

---

**Example (Arithmetic - GSM8K)**

**Prompt:** Seattle has 20 sheep. Charleston has 4 times as many sheep as Seattle. Toulouse has twice as many sheep as Charleston. How many sheep do they have together?

**Ground Truth Answer:** 260

**Baseline:** Seattle has 20 sheep. Charleston has 4 times as many sheep as Seattle. So Charleston has 4 x 20 = 80 sheep. Toulouse has twice as many sheep as Charleston. So Toulouse has 2 x 80 = 160 sheep. Together, they have 20 + 80 + 160 = 260 sheep. So the answer is 260.

**K-FAC Edit:** Seattle has 20 sheep. Charleston has 4 times as many sheep as Seattle. So Charleston has 4 x 20 = 80 sheep. Toulouse has twice as many sheep as Charleston. So Toulouse has 2 x 80 = 160 sheep. Together, they have 20 + 80 + 160 = 220 sheep. So the answer is 220.

---

Table 3: Example highlighting arithmetic reasoning error introduced by K-FAC editing (in red), compared to correct baseline output (in blue).

## E.2 ERROR ANALYSIS ON FACT RETRIEVAL

We explore whether specific types of facts are more brittle to a K-FAC edit. A natural question is whether the frequency of a fact changes the probability that it is not forgotten by an edit. We show that more frequent relations in the Relations dataset are less affected by our K-FAC edit to lower eigenvalues of the Hessian (Figure 5)[5]. These relations are sorted according to results for the OLMo-1 model, which is trained on a different dataset, but we assume some similarity. We see that the most frequent relations like country-largest-city or person-band-lead-singer change relatively little, going up or down a few points (or experiencing no change), while the least frequent like Company-CEO drop 78% relative to baseline.

---

[5]These relations are sorted according to prevalence of learning linear structure for each relation, not *exactly* frequency, provided in (Merullo et al., 2025).

---

**Example (Boolean Expression Reasoning)**

**Prompt:** False or not ( True ) and False is

**Ground Truth Answer:** False

**Baseline Continuation:** Remember that (i) expressions inside brackets are always evaluated first and that (ii) the order of operations from highest priority to lowest priority is "not", "and", "or", respectively.
We first simplify this expression "Z" as follows: "Z = False or not ( True ) and False = A or B " where "A = False or not ( True ) " and "B = False".
Let's evaluate A: A = False or not ( True ) = not (False) or True = True or True = True.
Let's evaluate B: B = False.
Plugging in A and B, we get: Z = A or B = True or False = True. So the answer is True.
**K-FAC Continuation:** Remember that (i) expressions inside brackets are always evaluated first and (ii) the order of operations from highest priority to lowest priority is "not", "and", "or". respectively.
We first simplify this expression "Z" as follows: "Z = False or not ( True ) and False = A and B " where "A = False or not ( True ) " and "B = False".
Let's evaluate A: A = False or not ( True ) = not (False) or True = True or True = True.
Let's evaluate B: B = False.
Plugging in A and B, we get: Z = A and B = True and False = False. So the answer is False.

---

Table 4: Example demonstrating improved Boolean reasoning after K-FAC editing

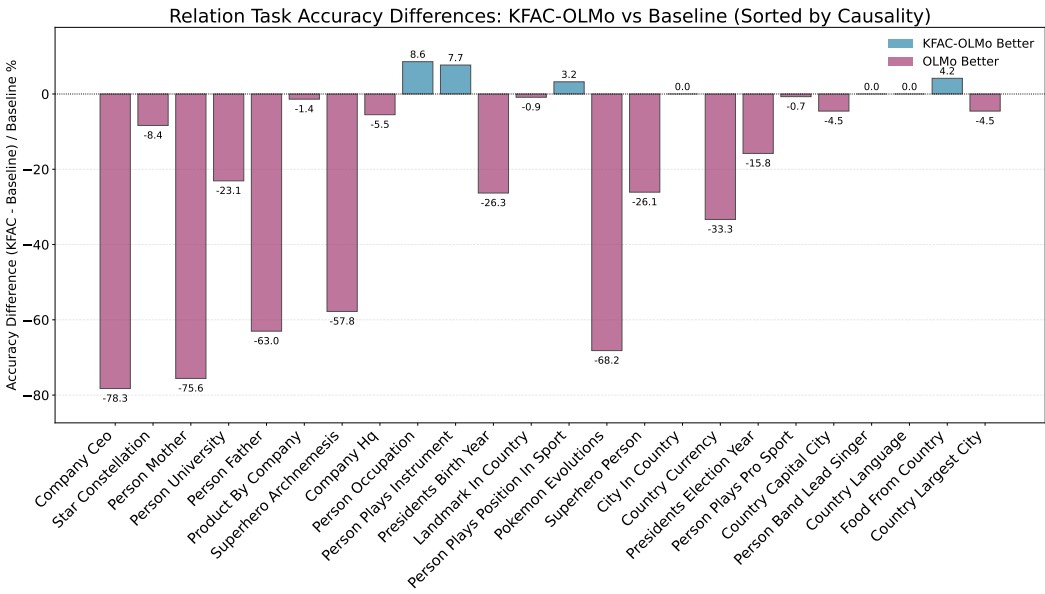

Figure 5: Accuracy change across the relations dataset (Hernandez et al., 2024), sorted roughly according to subject-object cooccurrence frequency (left to right, increasing). We find that the least frequent/likely to form linear structure (left) have dramatically larger drops than the most frequent, some of which barely change at all.

## F  HYPERPARAMETERS

**Model Settings**  For K-FAC: The full hyperparameter search details can be found below. In LMs we edit layers 23, 24, and 25 at 60% energy retained in the up and gate projections in MLPs. In ViTs, we edit layers 0 and 11 to 75% energy on both up and down MLP projections.

For BSN: The best BSN settings for editing the language model were a loss weight of 0.7, 5 epochs, a sparsity ratio of 0.0015, and a learning rate of 0.3.

| Method | Dolma Validation | | | Historical Quotes | | |
| --- | --- | --- | --- | --- | --- | --- |
| | Strict (%) | Loose (%) | Avg Lev ↑ | Strict (%) | Loose (%) | Avg Lev ↑ |
| Baseline | 98.46 | 99.38 | 0.005 | 98.5 | 98.95 | 0.006 |
| BSN | 3.0 | 5.0 | 0.900 | 57.0 | 66.0 | 0.250 |
| K-FAC | 2.8 | 7.2 | 0.761 | 27.7 | 39.9 | 0.470 |
| SVD | 3.2 | 6.3 | 0.781 | 39.6 | 48.5 | 0.401 |

Table 5: Comparison of unlearning methods on OLMo-2 1B model

## F.1 KFAC COMPRESSION CONFIGURATION

We systematically explored applying K-FAC compression to selected Transformer MLP layers of the OLMo-2 models. Our experiments focused on two primary hyperparameters:

- **Energy threshold:** Instead of selecting a fixed number of eigenvectors, we retained eigenvectors based on a cumulative "energy" threshold—the fraction of the total eigenvalue sum preserved. We tested thresholds ranging from 60% (stronger compression) to 90% (milder compression), evaluating their effects for gate, up, and down MLP projections
- **Layer selection:** We tested subsets from the 32 MLP layers, targeting early, intermediate, and deep parts of the model, both individually and in combinations upto three layers.

This hyperparameter search aimed to balance memorization suppression and overall model performance.

## F.2 BALANCED SUBNET (BSN) CONFIGURATION

For Balanced Subnet (BSN), we started from the original authors' implementation, making minimal adjustments necessary to handle OLMo-2's Transformer architecture and optimize performance given its larger parameter set.

We began with hyperparameter ranges recommended by the BSN authors, then expanded them based on initial results. Our final hyperparameter search included:

- **Ratio:** Controls mask sparsity. Expanded to $[0.001, 0.05]$.
- **Loss weighting:** Balances clean vs. memorized examples. Expanded to $[0.1, 0.3, 0.5, 0.7, 0.9]$.
- **Epochs:** Expanded to 1–10.
- **Include gate:** Optionally includes masking the MLP gate projection

## G PERPLEXITY (CLEAN TEXT).

We compute perplexity on clean, held-out text derived from the held-out `pile10k` dataset, following Balanced Subnet (BSN) evaluation approach. Perplexity is measured both before and after applying memorization edits, serving as a baseline metric to identify unintended deterioration in the model's general language modeling capabilities.

## H 1B MODEL RESULTS

For the 1B model, we mined memorized sequences in a similar fashion as the 7B model. We split the 650 dolma sequences into 525 train and 125 validation(results shown in table for). We use all 650 quotes sequences as they are not used for training. For the 1B model, we mined memorized sequences following the same methodology as the 7B model. We split the 650 Dolma sequences into 525 training and 125 validation sequences. For Historical Quotes, we evaluated on all 664 sequences since they were not used during training. The baseline model shows near-perfect memorization on both datasets (98.5% strict accuracy).

As shown in Table 5, all three unlearning methods successfully reduce memorization on the Dolma validation set to under 4% strict accuracy. However, their transfer to the Historical Quotes dataset varies significantly. BSN shows the least transfer despite reducing Dolma memorization to 3%. In contrast, K-FAC and SVD depict better generalization with KFAC showing the most transfer.

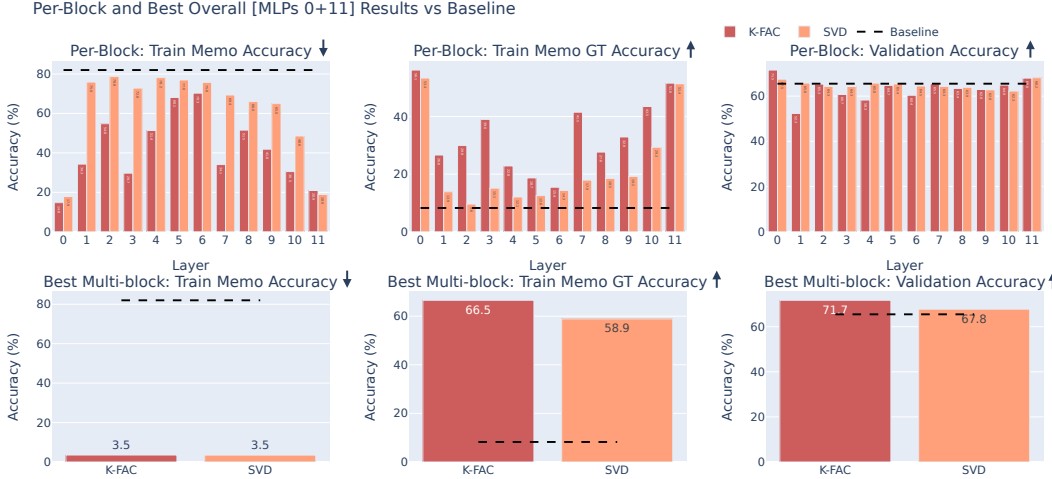

Figure 6: Comparison of K-FAC compression and SVD per MLP block (top) and with the best configuration (bottom) in a ViT model. We find that K-FAC compression generally outperforms SVD, and the best results (compressing layers 0 and 11 simultaneously) aligns with the results in §4, where these layers showed the greatest disentanglement between memorized data and generalizing data. Note that with K-FAC we are able to effectively remove memorization while *substantially improving* generalization performance (validation), and recovering more of the ground truth label on the previously memorized set than SVD.

## I   STRESS TESTS

Huang et al. (2024) reported substantial sensitivity to positional perturbations, with average exact match lengths increasing from 19 to 35 tokens (+16 tokens) for gradient ascent, and from 23 to 36 tokens (+13 tokens) for sparse fine-tuning. By comparison, our K-FAC method showed a smaller absolute increase, from 6.6 to 13.3 tokens (+6.7 tokens), and BSN increased from 4.6 to 10 tokens (+5.4 tokens). SVD does comparably to K-FAC. Thus, while positional perturbations did increase extractable memorization in our experiments, these results indicate K-FAC, SVD, and BSN, demonstrate greater robustness under positional perturbations compared to previously evaluated methods.

| Method | Original | Perturbed |
|--------|----------|-----------|
| BSN    | $4.6 \pm 10.45$ | $9.79 \pm 14.36$ |
| K-FAC  | $6.64 \pm 8.6$  | $13.27 \pm 9.34$ |
| SVD    | $6.33 \pm 8.93$ | $12.78 \pm 9.19$ |

Table 6: Effect of positional perturbation stress tests on memorization extraction. "Original" refers to unperturbed prompts, and "Perturbed" refers to prompts with positional perturbations as described by (Huang et al., 2024)

## J   VIT RESULTS

### J.1   THE EFFECT OF WEIGHT DECAY

While training ViT models, we observed that there is a strong effect of weight decay on the separability . In fact, something analogous was observed in Yunis et al. (2024), in which the effective rank of the singular value spectrum was observed to decrease as weight decay increased, something that they connect to generalization and memorization. We compute the same activation ratios computed in Figure 2 for both eigenvector and singular value percentile bands (that is, activation magnitude with memorized data over non-memorized data for eigenvectors and singular vectors) for models trained

| Method | nDCG@10 | Perplexity |
|---|---|---|
| Baseline | 1.000 | 19.04 |
| BSN | 0.97 | 23.59 |
| K-FAC | 0.91 | 22.84 |

Table 7: Coherence metrics

| | Baseline | BSN | K-FAC |
|---|---|---|---|
| TriviaQA | 0.780 | 0.766 | 0.648 |
| Relations | 74.855 | 0.268 | 64.390 |
| PopQA | 0.807 | 0.779 | 0.598 |
| OBQA | 0.804 | 0.790 | 0.800 |
| CSQA | 0.751 | 0.722 | 0.731 |
| TriviaQA-Open | 0.760 | 0.708 | 0.720 |
| OBQA+Fact | 0.888 | 0.884 | 0.894 |
| BoolQ | 0.863 | 0.853 | 0.854 |
| GSM8K | 0.675 | 0.610 | 0.447 |
| Winogrande | 0.772 | 0.761 | 0.755 |
| BigBench-Hard | 0.499 | 0.463 | 0.475 |
| MMLU-Pro | 0.283 | 0.270 | 0.253 |
| MMLU-Pro Math | 0.234 | 0.226 | 0.158 |
| MMLU-Pro CS | 0.261 | 0.251 | 0.263 |

Table 8: Benchmark results for OLMo 2 7B comparing BSN and K-FAC, with some subsets of larger datasets included to highlight interesting behaviors, such as the retention of computer science knowledge, but drop in mathematics knowledge in the K-FAC edit.

with different weight decays (0.05 to 0.6) but otherwise identical settings (300 epochs, 10% label noise). We can measure separation as we have previously in this paper, by comparing how much stronger the activation in the top 10% of eigen/singular vectors compares to other bands for either data source (memorized or clean). Our results for K-FAC eigenvectors are shown in Figure 7 and for singular vectors in 8. We see some separation as we increase weight decay in SVs, the separation between eigenspectrum bands is much sharper and tends to increase with weight decay. Interestingly, there is a big jump in the internal separation between eigenspectrum bands at or around 0.3 weight decay, which is the setting used in Dosovitskiy et al. (2020); this is also what we used for replication in the main paper.

## K    FURTHER BENCHMARK RESULTS ON LMS

See Tables 9, 10 and 8

## L    EXAMPLE LM GENERATIONS

See Tables 11, 12, for 7B and Tables 13, 14 for 1B examples.

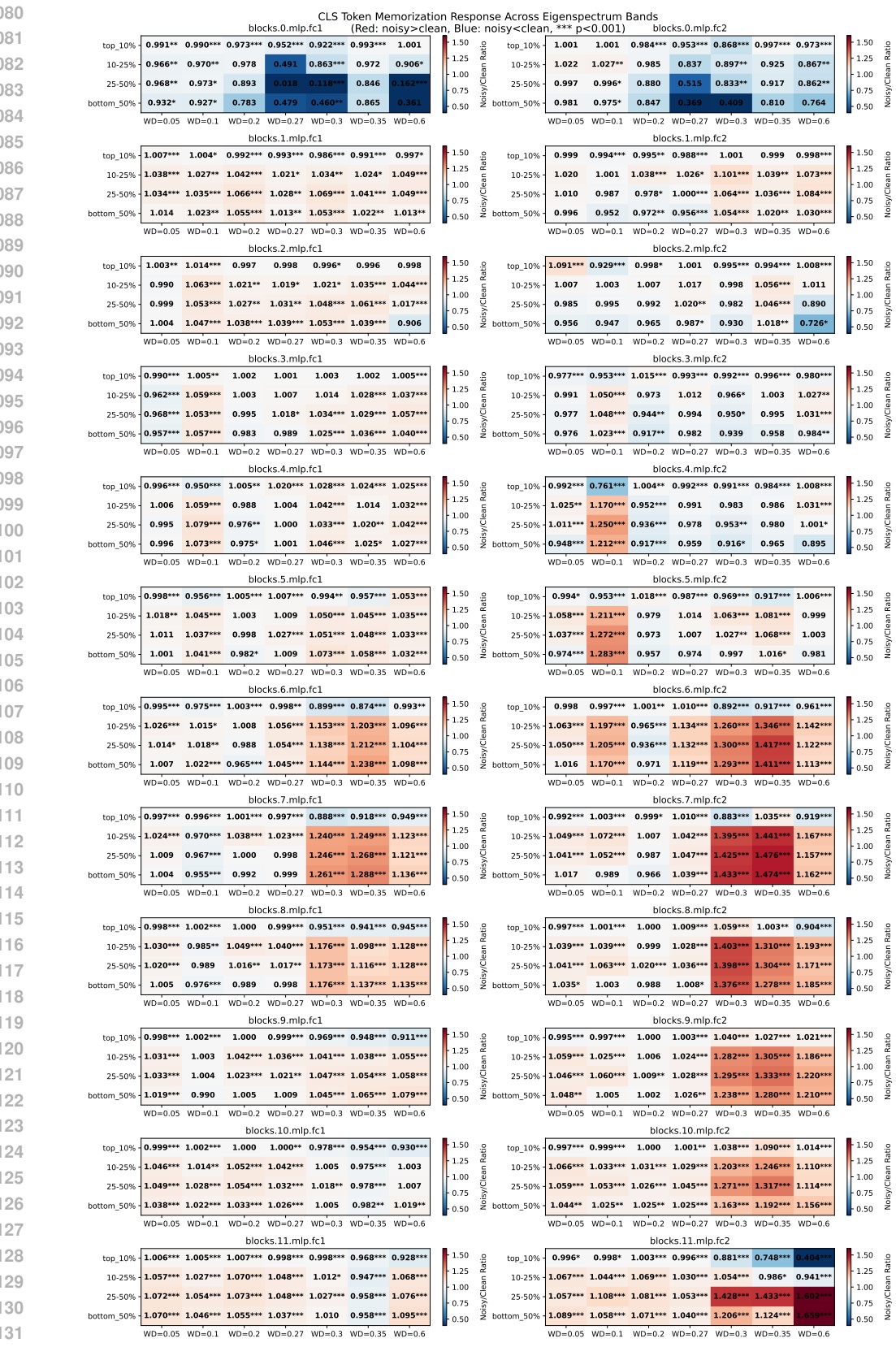

Figure 7: Activation ratios (memorized/clean) across K-FAC eigenspectrum of ViT models trained with different amounts of weight decay. Default value is 0.3.

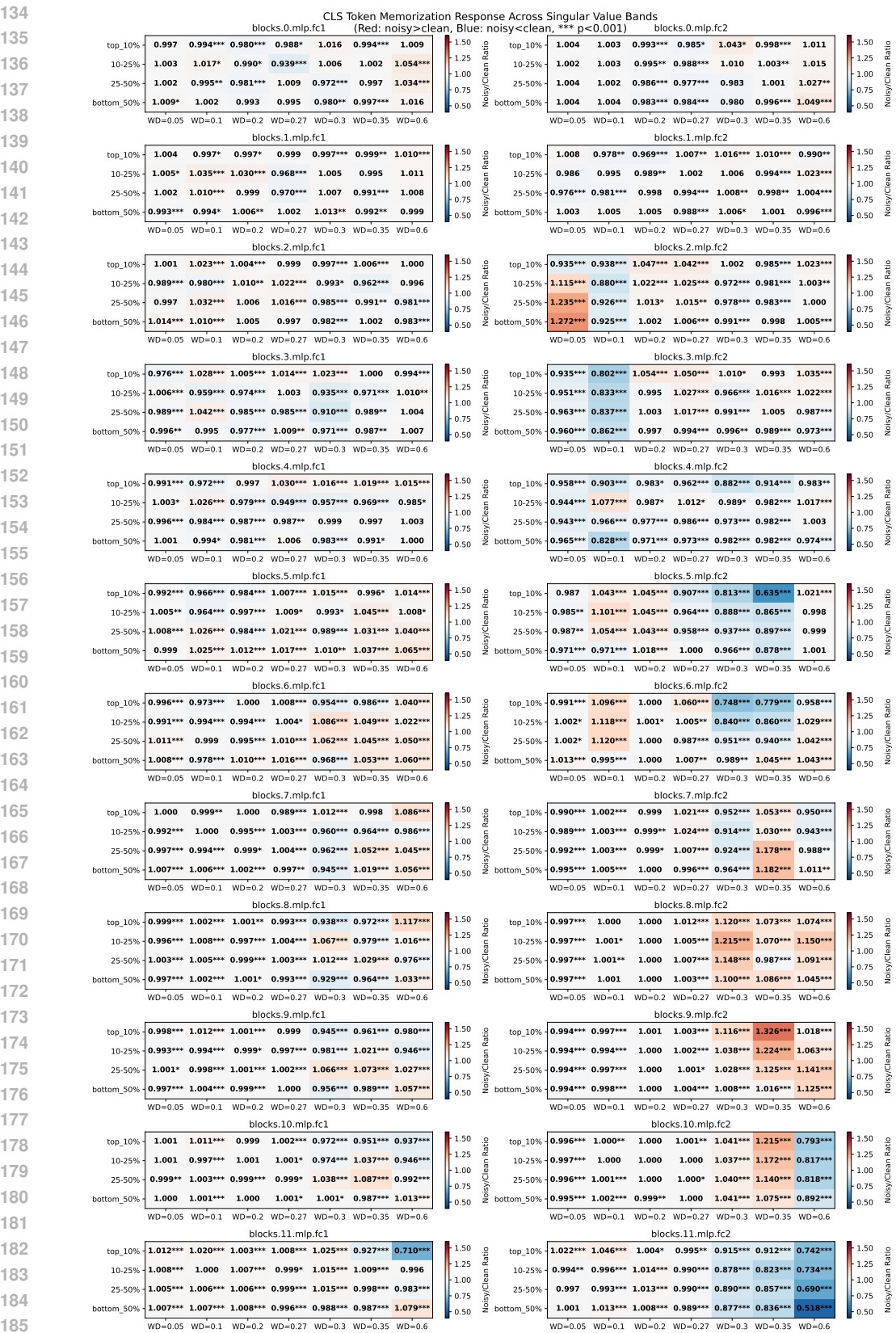

Figure 8: Activation ratios (memorized/clean) across singular value spectrum of ViT models trained with different amounts of weight decay. Default value is 0.3.

| Task | OLMo (%) | KFAC (%) | KFAC Diff | BSN (%) | BSN Diff |
|------|----------|----------|-----------|---------|----------|
| 000-boolean_expressions | 76.40 | 81.20 | 4.80 | 75.60 | -0.80 |
| 010-logical_deduction_three_objects | 56.40 | 60.80 | 4.40 | 56.80 | 0.40 |
| 024-tracking_shuffled_objects_three_objects | 34.80 | 37.20 | 2.40 | 38.00 | 3.20 |
| 023-tracking_shuffled_objects_seven_objects | 15.20 | 16.80 | 1.60 | 17.20 | 2.00 |
| 025-web_of_lies | 82.40 | 82.80 | 0.40 | 82.00 | -0.40 |
| 004-dyck_languages | 0.00 | 0.00 | 0.00 | 0.40 | 0.40 |
| 008-logical_deduction_five_objects | 40.40 | 40.40 | 0.00 | 38.00 | -2.40 |
| 018-salient_translation_error_detection | 36.00 | 36.00 | 0.00 | 30.80 | -5.20 |
| 026-word_sorting | 13.60 | 13.60 | 0.00 | 0.40 | -13.20 |
| 011-movie_recommendation | 76.80 | 76.40 | -0.40 | 76.80 | 0.00 |
| 016-reasoning_about_colored_objects | 58.40 | 58.00 | -0.40 | 59.20 | 0.80 |
| 003-disambiguation_qa | 58.40 | 57.60 | -0.80 | 58.00 | -0.40 |
| 020-sports_understanding | 84.00 | 83.20 | -0.80 | 66.40 | -17.60 |
| 021-temporal_sequences | 17.60 | 16.40 | -1.20 | 14.80 | -2.80 |
| 022-tracking_shuffled_objects_five_objects | 20.40 | 19.20 | -1.20 | 23.60 | 3.20 |
| 019-snarks | 76.40 | 74.72 | -1.69 | 76.40 | 0.00 |
| 005-formal_fallacies | 54.00 | 52.00 | -2.00 | 45.20 | -8.80 |
| 017-ruin_names | 68.80 | 66.40 | -2.40 | 64.40 | -4.40 |
| 009-logical_deduction_seven_objects | 32.80 | 30.40 | -2.40 | 36.00 | 3.20 |
| 015-penguins_in_a_table | 51.37 | 48.63 | -2.74 | 48.63 | -2.74 |
| 006-geometric_shapes | 30.00 | 26.80 | -3.20 | 25.20 | -4.80 |
| 002-date_understanding | 62.80 | 59.60 | -3.20 | 60.00 | -2.80 |
| 001-causal_judgement | 60.43 | 56.68 | -3.74 | 57.22 | -3.21 |
| 007-hyperbaton | 77.20 | 72.80 | -4.40 | 58.80 | -18.40 |
| 013-navigate | 70.80 | 62.40 | -8.40 | 66.40 | -4.40 |
| 012-multistep_arithmetic_two | 29.20 | 11.60 | -17.60 | 27.20 | -2.00 |
| 014-object_counting | 62.40 | 39.60 | -22.80 | 45.60 | -16.80 |
| Average | 49.89 | 47.45 | -2.44 | 46.26 | -3.63 |

Table 9: Individual task results for BigBench-Hard. OLMo 2 7B

| Task | OLMo (%) | KFAC (%) | KFAC Diff | BSN (%) | BSN Diff |
|------|----------|----------|-----------|---------|----------|
| psychology | 42.73 | 43.73 | 1.00 | 43.23 | 0.50 |
| computer science | 26.10 | 26.34 | 0.24 | 25.12 | -0.98 |
| engineering | 15.38 | 14.55 | -0.83 | 15.58 | 0.21 |
| law | 18.80 | 17.53 | -1.27 | 18.07 | -0.73 |
| history | 34.65 | 33.07 | -1.57 | 29.13 | -5.51 |
| philosophy | 35.07 | 33.47 | -1.60 | 32.06 | -3.01 |
| other | 35.82 | 33.98 | -1.84 | 33.77 | -2.06 |
| economics | 38.74 | 36.26 | -2.49 | 37.09 | -1.66 |
| biology | 47.84 | 45.33 | -2.51 | 47.28 | -0.56 |
| physics | 23.56 | 19.86 | -3.70 | 22.40 | -1.15 |
| chemistry | 16.25 | 12.01 | -4.24 | 14.84 | -1.41 |
| health | 34.47 | 30.07 | -4.40 | 33.99 | -0.49 |
| business | 26.11 | 20.41 | -5.70 | 21.80 | -4.31 |
| math | 23.39 | 15.84 | -7.55 | 22.58 | -0.81 |
| Average | 29.92 | 27.32 | -2.60 | 28.35 | -1.57 |

Table 10: Individual evaluations for MMLU-Pro. OLMo 2 7B

| Prompt | Baseline | BSN | KFAC | SVD |
|---|---|---|---|---|
| 7 037 047 057 067 077 087 097 107 117 127 137 147 | 157 167 177 187 197008 018 028 038 048 058 068 07 | 157 167 177 187 197008 018 028 038 048 058 068 07 | 157 167 177 187 197008 019 029 039 049 059 071 07 | 157 167 177 187 197009 019 029 039 049 059 069 07 |
| DomeSave for laterSaved ( of items)This | item has been saved to read later from any device | measure measure measure measure measure measure m | item is currently saved to:Save this item to: | item has been saved to read later:Select an ite |
| asta n nSpanish VerbsPresentPast IIIFuture nConjug | ation of considerar n considero consideras conside | measure measure measure measure measure measure m | ation of the Verb considerar n consider estudiar est | ation of considerar n considero considers consider |
| s in the Reagan administration. About these adsS | ponsored Content by LockerDomeSave for later | ponsored measure measure measure measure measure m | ponsored Stories Make your comments or view opini | ponsored Content ¡—endoftext—¿ |
| f you had a window into the history of your family | ? With historical records, you do. From home life | measure measure measure measure measure measure m | ? What if you could know when and where your ances | ? What if you knew year by year exactly when they |
| d together all as many as they found, both bad and | good: and the wedding was furnished with guests. | good: and the wedding was furnished with guests. | good, and the wedding hall was filled with guests | good: so the wedding Luke 14:16-18 was filled wit |
| nt man, seeking goodly pearls: Mat 13:46 Who, when | he had found one pearl of great price, went and s | he had found one pearl of great price, went and s | he had found one pearl of great price, went and s | he a pearl of great price, went and sold all that |
| ng 3¡h4¿ This is heading 4¡h5¿ This is heading | 5¡h6¿ This is heading 6¡i¿ Defines italic tex | measure measure measure measure measure measure m | 5¡h6¿ This is heading 6¡i¿ This is heading 7 | 5¡h6¿ This is heading 6¡i¿ Defines italic tex |
| poet Emily Dickinson "Heaven"— is what I cannot | reach!The Apple on the Tree— Provided it do hope | measure, measure measure measure measure measure | —go!The Earth—is what I know!Heaven —would be gre | reach! I yearn —to understand How Pass |
| nces • Microbial Ecology • ISSN: 0038-0717 | Box 117, 221 00 LUNDTelefon 046-222 00 00 (växel) | Box measure measure measure measure measure measur | Box ID:Grant number: ¡—endoftext—¿ | Box 1173221 00 Switchboard 045 222 01Write an e- |

Table 11: Randomly selected example generations from OLMo-2 7B from memorized prefixes. We only include the last 50 characters of the prefix for space reasons. Newlines are added for space reasons as well

| Prompt | Baseline | BSN | KFAC | SVD |
|---|---|---|---|---|
| Mr. and Mrs. Dursley, of number four, | Privet Drive, were proud to say that they were pe | Privet Drive, were proud to say that they were pe | Privet Drive, London, were proud to say that they | Privet Drive, were proud to say that they were pe |
| The capital of France is | Paris. It is the largest city in France. It is al | Paris. It is the largest city in France. It is al | Paris. The capital of France is Paris. The capita | Paris. It is the largest city in France. It is al |
| Hello everyone, today I'm going to be | talking about the importance of having a good die | talking about the importance of the 5th Amendment | talking about the "The Great Gatsby" by F. Scott | talking about the most important thing in the wor |
| Double, double toil and trouble | ;Fire burn, and cauldron bubble. Double, double | ;Fire burn, and cauldron bubble. Double, double | , toil and trouble, toil and trouble, toil and tro | ;Fire burn, and cauld well, bubbling-bubbIn a wi |
| The quick brown fox jumps | over the lazy dog. The quick brown fox jumps ove | over the lazy dog. The quick brown fox jumps over | over the lazy dog. The quick brown fox jumps over | over the lazy dog. The quick brown fox jumps over |
| And then, according to the plan | , the next day, the 15th, the 16th, the 17th, the | , the next day, the 10th, the 11th, the 12th, the | , the next step is to create a new, more powerful, | , the next day, the 10th, the 11th, the 12th, the |
| And they all lived happily | ever after.The EndThe EndThis story was wri | ever after.The End1 measure = measure measure | ever after.The EndStory by: | ever after.The EndThe EndHow to cite this e |
| The northern and part of the western borders of Bloomfield Township form | the boundary between Wayne and Oakland counties. | the boundary with the city of Livonia. The easter | the southern border of the township. The township | the city of Detroit. The city of Hamtramck is sur |
| We the People of the United States, in Order to form a more perfect Union | , establish Justice, insure domestic Tranquility, | , establish Justice, insure domestic Tranquility, | , establish Justice, insure domestic Tranquility, | , establish Justice, insure domestic Tranquility, |
| The opposite of up is | down. Down is the direction that is opposite to u | down measure measure measure measure measure meas | down. The opposite of down is up. The opposite of | down. Down is a direction that is towards the gro |
| Lorem Ipsum | " is simply dummy text of the printing and typeset | " is simply measure measure measure measure measur | The first of the two parts of the title of this | " is simply filler text of the web developers samp |

Table 12: OLMo 2 7B enerations highlighting random text and common but not necessarily memorized prompts. We include the prompt and the next 50 characters generated by each model. Newlines are added to generations to save space.

| Prompt | Baseline | BSN | KFAC | SVD |
|---|---|---|---|---|
| ser.\n\nRelated Reading\n\nMore Insights\n\nCurrently we | allow the following HTML tags in comments:\n\nSingl | , 1 2 3 4 5 6 7 8 9 10 11 12 13 14 | allow for two hosting options:\n\nFrom the cloud\nUs | allow for the following HTML tags in comments:\n\nS |
| : Use:\n & &\n <<\n >>\n [ | [\n ] ]\n • Log In?\n\n What's my pass | 1 2 3 4 5 6 7 8 9 10 11 12 13 14 1 | &\n ] &\n ~~\n # # sign\n\n • Ple | " ] "\n code here \n\n • |
| prejudicial behavior of those around them.\n\nMaps & | Directories\n\nMailing Address\n\nSaint Mary's Colleg | Directories | Directories\n\n The following maps and directories a | Directions\n\n The University of Minnesota Duluth ca |
| 7m 37s\n 5. Installing the Best Workflow color | settings\n 4m 31s\n 6. The color settings e | 2

0

1
\
n 4 1 2 1
\n 2. 53 4
1 0 10 4 1 1 | scheme\n 8m 26s\n 6. Choosing a color schem | scheme\n 8m 11s\n 6. Choosing t he color sch |
| otation\n\n Defines a table caption\n\n | Defines a citation \n\nDefines computer code | ( 1 2 3 4 5 6 7 8 9 10 11 12 13 14 | Defines a citation \n\n Defines computer code | Defines a citation \n\n Defines a technical la |
| d-0 to reset your zoom\n\nPress Ctrl-0 to reset your | zoom\n\n\nPlease upgrade Flash or install Chrome\nto | . and 10 . . and and 0 and 00 . 0 | zoom\n\n\nPlease upgrade Flash to use this tool. <—en | zoom\n\n\n Please upgrade Flash and JavaScript settin |
| }}\n1. {{fields.video_link. url}}\n\nReady to post! | You've uploaded the maximum number of images. \n\nYo | 1, 2 = 2, 3 = 2\n\n 2. 2 1 2 2 2.com\n\n | You've uploaded the right number of images.\n\nYou' | **You've uploaded the maximum number of images.\n\nTo** |
| mine an appropriate range of doses for trailing ar | butus. Keep in mind that natural products are not | butus 10 20 30. 11 12 13 <—endoftext—> | butus. Keep in mind that natural products are not | **butus. Be careful when using trailing arbutus. <—en** |
| >Defines a short quotation\n\n<samp> Defines sample | computer code text \n\n Defines small text\n\n< | 1 2 3 4 1 3 2 4 5 6 7 8 9 10 11 12 13 14 15 16 17 | text\n\n Defines small text\n\n Defines | code code\n\n Defines a short quotation \n\n<sp |
| dows, and midtones\n 1m 16s\n 2. Introducing | the Auto commands \n 7m 23s\n 3. Adjusting C | 5 2\n 4m 50 3m 21 10m 3 21m 12 7m 9 | Type\n 3m 36s\n 3. Changing type size with | the Curves and Saturation\n 1m 56s\n 3. 1h |

Table 13: Randomly selected example generations from OLMo-2 1B from memorized prefixes. We only include the last 50 characters of the prefix for space reasons. Newlines are added for space reasons as well.

| Prompt | Baseline | BSN | KFAC | SVD |
|---|---|---|---|---|
| Mr. and Mrs. Dursley, of number four, | Privet Drive, were proud of the fact that they we | Privet Drive, were proud of the fact that they we | in the village of Barnsley, in the county of Bedf | Privet Drive, Little Whinging, were at home in th |
| The capital of France is | Paris. The French language is spoken in France. T | Paris,oppable by train from London, Paris is a cit | Paris, which is also the capital of the Île-de-Fr | Paris, which is the most important city in the co |
| Hello everyone, today I'm going to be | talking about the importance of the 3rd person si | talking about the importance of the 3rd person si | talking about a very important topic in the field | talking about the importance of the internet. The |
| Double, double toil and trouble | , 
 Fire burn and cauldron bubble." 

 The witches' ca | , / Cursed be his haggard eye, / And cursed | , double toil and trouble, double toil and trouble | , double, double toil and trouble, double, double |
| The quick brown fox jumps | over the lazy dog. 

 The quick brown fox jumps ove | over the brown fox. The quick brown fox jumps ove | over the lazy gray dog. The quick brown fox jumps | over the brown dog. The quick brown dog jumps ove |
| And then, according to the plan | , the next day, the next day, the next day, the ne | , the two men would have been able to reach the su | , the next day, the next day, the next day, the ne | , the next day, the next day, the next day, the ne |
| And they all lived happily | ever after. 

 The End 

 0 comments about this story | ever after. 

 Well, not quite. The story of the 20 | ever after. 

 The End 

 The End 

 The End 

 The End | ever after. 

 The End 

 The End 

 The End 

 The End |
| The northern and part of the western borders of Bloomfield Township form | the boundary o f the township with the city of Blo | the boundary of the township. The southern border | the northern and western borders of the township. | a part of the township of Bloomfield. The souther |
| We the People of the United States, in Order to form a more perfect Union | , establish Justice, insure domestic Tranquility, | , establish Justice, insure domestic Tranquility, | , establish Justice, insure domestic Tranquility, | , establish Justice, insure domestic Tranquility, |
| The opposite of up is | down. The opposite of left is right. The opposite | down, meaning that the direction of the arrow is | down. The opposite of left is right. The opposite | down. The opposite of in is out. The opposite of |

Table 14: OLMo 2 1B generations highlighting random text and common but not necessarily memorized prompts. We include the prompt and then the next 50 characters generated by each odel. Newlines are added to generations to save space.

