# OpenReview forum: "From Memorization to Reasoning in the Spectrum of Loss Curvature"
_ICLR.cc/2026/Conference — Submitted to ICLR 2026_

### Official Review · Reviewer_UU7q · 2025-10-22

**Soundness:** 2
**Presentation:** 1
**Contribution:** 2
**Rating:** 2
**Confidence:** 3

**Summary:**

This paper investigates how memorization and generalization are represented in Transformers by analyzing loss curvature. Using the K-FAC approximation of the Hessian, the authors argue that high-curvature directions correspond to shared, generalizable structures, while low-curvature directions relate to memorization and brittle behaviors. Based on this finding, they propose a weight editing method that selectively removes low-curvature components to suppress memorization while preserving model performance. The authors evaluate their approach on both language models and vision transformers, comparing against the BalancedSubnet (BSN) baseline. They additionally find that different tasks exhibit varying sensitivity to the removal of low-curvature directions.

**Strengths:**

The paper offers a fresh perspective on the memorization-generalization trade-off through the lens of loss curvature. The investigation of how curvature relates to different cognitive abilities is interesting and raises fundamental questions about the geometry of neural loss landscapes.

**Weaknesses:**

While the results are promising, I believe the paper requires a significant update before being accepted. I detail the main reasons for that below.

- **Limited scope of curvature analysis.** The reliance on K-FAC applied only to MLP weight matrices, while computationally practical, undermines the broader claims. The authors argue that curvature affects generalization and reasoning, but K-FAC analysis restricted to MLP weights lacks the precision needed to support such claims. Notably, [prior work](https://proceedings.mlr.press/v162/benzing22a/benzing22a.pdf) has demonstrated that K-FAC does not necessarily capture true second-order information. Moreover, the Fisher Information Matrix (and its K-FAC approximation) represents only one component of the Hessian, providing an incomplete picture of curvature.
- **Lack of precision of the proposed editing method.** While the proposed approach effectively reduces certain forms of memorization, it appears to damage adjacent information that should be preserved. For instance, memorization of historical quotations represents valuable knowledge that the method inadvertently destroys. The performance degradation on reasoning tasks (Figure 2) is non-negligible. The BSN method appears more targeted in its editing; I suspect an analogous Figure 2 for BSN would demonstrate significantly better preservation of reasoning abilities. Additionally, the paper focuses on knowledge and reasoning tasks but leaves unclear whether other capabilities are compromised by the editing process.
- **The paper is somewhat hard to read.** The high level structure makes a lot of sense, but individual sections are difficult to follow. The logical flow of arguments and the methodological details often lack clarity, making it challenging to understand precisely what the authors are doing.

**Questions:**

- In Figure 1, could the average norm of the vectors affect the overall shape of the evolution? In particular, could it explain the big change in ratio in the last layer of ViTs?
- Is there a specific reason on why keeping other types of parameters (e.g. attention weight matrices) outside the scope of the analysis?

---

> ### Public Comment · ~Andy_Barcia-Rodríguez1 · 2025-11-14
>
> Disclaimer: I am not an author or reviewer of this paper, just an external reader following this thread.
>
> I wanted to offer a different perspective on one of the points raised in this review, specifically:
>
> > "memorization of historical quotations represents valuable knowledge that the method inadvertently destroys."
>
> Whether verbatim memorization of such content should be preserved seems, to me, more like a design choice about what kind of model we want, rather than a  drawback. There is a growing line of thought that models may move toward a "cognitive core", prioritizing general reasoning and abstraction over storing encyclopedic or verbatim knowledge. Under that lens, a method that preferentially suppresses memorization (e.g., of exact quotations) while largely preserving performance on more general tasks could be viewed as a feature rather than a flaw.

---

> ### Author Response · Authors · 2025-12-04
>
> Thank you to the reviewer for their review and comments, but we believe there are a few misconceptions about the work, perhaps because the submission version was a bit hard to read.
>
> > The paper is somewhat hard to read.
>
> We apologize for this. We have made many updates to the revision version to make it more readable.
>
> > Limited scope of curvature analysis.
>
> K-FAC is indeed an approximation. If it was not a good enough approximation, we would not see such positive results that also align so well with background work on memorization and curvature. That does not mean our edit is perfect. As we report, we see some hit to perplexity.
>
> >  it appears to damage adjacent information that should be preserved. For instance, memorization of historical quotations represents valuable knowledge that the method inadvertently destroys
>
> The point of our method is to see what happens when we target memorization *globally*, not to compete with a targeted method like BSN. We agree with the public comment on this
>
> > Is there a specific reason on why keeping other types of parameters (e.g. attention weight matrices) outside the scope of the analysis?
>
> A point about MLPs/Attention: MLP weights are about 66% of model parameters, replicable prior work connects MLP weights to memorization/fact retrieval. The only part of attention weights that may contain memorized information would be the Value and Output matrices. Given that these are such a small part of the overall parameter count, we consider it acceptable to forgo using them in our analysis for simplicity.
>
>
> > the paper focuses on knowledge and reasoning tasks but leaves unclear whether other capabilities are compromised by the editing process.
>
> Which other capabilities? We look at over 60 individual evaluations spanning multiple capabilities. Without specifics we can not address this point.

---

### Official Review · Reviewer_p2VL · 2025-10-29

**Soundness:** 3
**Presentation:** 4
**Contribution:** 3
**Rating:** 4
**Confidence:** 3

**Summary:**

This paper studies memorization in Transformer models. It shows that the eigenbasis of the approximated Hessian of weight matrices uncovers distinct disentanglement of memorization and generalization, across vision and language models. In a population-level of samples, memorization directions are actually flatter than generalizing directions. They then propose a recitation-reduction technique based on ablating memorized directions in weight space.

**Strengths:**

- The paper is clearly written and easy to read.

- The use of K-FAC to analyze memorization and generalization is novel to me.

- The results that memorization aggregated over a population level is actually flatter than generalization direction are interesting.

**Weaknesses:**

1. The limitation is not well discussed. For example, in Table 5, BSN is able to preserve higher accuracy than K-FAC edit. Also, the method requires a sweep to find which layers to edit, which is not well acknowledged as a limitation I believe. Are there other relevant limitations? For example, how does the computational cost of the proposed method compare with BSN?

2. While Line 143-149 provides a discussion, there could have been more analysis on per-example curvature analyses and population-level analyses. For example, showing how per-sample memorization directions get canceled out when averaging across samples in an experimental way would be nice.

**Questions:**

1. Can you clarify why you (only) choose BSN for comparison? Is BSN the only suitable baseline?

2. Can you clarify the relation between C and Z? Also, at Line 157, C is a rank-one component, but at Line 272, C seems not?

3. Is it a fair comparison when you need to sweep to get the layers to be editted and use different energy threshold in different cases? And how are the energy threshold selected?

I would be happy to increase score if the author can address some of these questions and weaknesses.

---

> ### Author Response · Authors · 2025-12-04
>
> We’d like to thank the reviewer for their comments. We are glad that the reviewer found the connection between generalization and curvature interesting and the work novel. We also think the reviewer makes important points that we would like to address here.
>
>
> > Can you clarify why you (only) choose BSN for comparison? Is BSN the only suitable baseline?
>
> This is a very important point. The original BSN paper (https://arxiv.org/abs/2410.02159), shows that BSN far outperforms existing methods by quite a large margin. Based on insights from Huang et al., 2024, we also concluded that other methods are highly susceptible to breaking down in stress tests that reveal they don’t remove much memorization. Therefore, for simplicity, we decided to only focus on the strongest possible baseline, which is BSN.
>
>
> > Can you clarify the relation between C and Z? Also, at Line 157, C is a rank-one component, but at Line 272, C seems not?
>
> This was terminology overload and we fixed this. Your interpretation is correct,
>
> > The limitation is not well discussed
>
> Apologies that the reviewer does not believe these are properly addressed. The sweep is not a limitation, but a part of the method, it uses the same amount of “supervision” as BSN. **However** to address this, we demonstrate that our method also supports an **entirely unsupervised approach** which BSN can not do. Please see the general comment.
>
> > Is it a fair comparison when you need to sweep to get the layers to be editted and use different energy threshold in different cases?
>
> It is a fair comparison because BSN is fully supervised. We simply use the same data to show that the mechanistic signal we are targeting does in fact affect memorization as well or better when provided the same supervision.
>
> > And how are the energy threshold selected?
> This is also swept.
>
>
> > While Line 143-149 provides a discussion, there could have been more analysis on per-example curvature analyses and population-level analyses. For example, showing how per-sample memorization directions get canceled out when averaging across samples in an experimental way would be nice.
>
> Thank you for raising this. We added a short experiment to demonstrate that this is indeed the case: lower eigenvectors correspond to a single (or small) set of memorized examples, and those directions are almost completely orthogonal to each other. Please see the general comment.
>
>
> We believe we have addressed the reviewer’s noted weaknesses, and we would like to thank them for raising salient points for us to address.

---

### Official Review · Reviewer_Qin4 · 2025-10-30

**Soundness:** 3
**Presentation:** 3
**Contribution:** 3
**Rating:** 6
**Confidence:** 3

**Summary:**

This paper proposes a method to edit model weights in Fisher curvature basis in order to tackle issues such as memorisation. Its central claim is that the Fisher Information Matrix of the model weights ($F_W$), when approximated layer-wise using the Kronecker-Factored Approximation (K-FAC), approximates the _curvature structure_ of the loss landscape. Each layer’s Fisher matrix is decomposed into Kronecker factors of input-activation and output-gradient covariances ($F_W \approx G \otimes A$), yielding an interpretable eigenstructure in which the curvature along a weight direction is given by $\lambda_i(G)\mu_j(A)$. High-curvature directions correspond to shared, generalisable mechanisms, while low-curvature modes that can be identified and pruned in the curvature basis, encode underconstrained, memorised artifacts whose removal reduces memorisation without degrading reasoning performance. Experiments are run on OLMO-2 7B with Balanced SubNet (BSN) as a baseline and on ViT-Base, on a variety of tasks, to confirm that pruning low-curvature Fisher modes reduces memorisation while preserving general reasoning and language quality.

**Strengths:**

- The work connects K-FAC to the semantic decomposition of model weights, being a novel application from its usage in optimisation. The semantic decomposition itself is shown to capture similar curvature-memorisation patterns in both language and vision, suggesting a a domain-agnostic property of deep networks.
- Reframes memorization vs generalization as spectral structure in curvature, bridging sharpness-aware optimization and localization studies.
- Unlike existing approaches like BSN which require labeled forget/retain sets, the K-FAC-based method requires no supervision beyond unlabeled data to compute the Fisher Information Matrix, in addition to memorisation labels across layers.
- The paper also makes some interesting connections with potential for subsequent research:
     - can yield an explanation for why low-curvature subspaces harbour privacy leakage and memorised facts,
     - by being tractable, the curvature basis can lend to specific weight editing or interpretability of model weights for concepts to be controlled for

**Weaknesses:**

- Since the layer-wise K-FAC blocks ignore cross-layer effects, can you comment on how relevant such cross-layer correlations and coupling between attention weights and the weights of a layer's MLP would be? Is this a limitation of the proposed approach, or it is not relevant?
 - How do we know that the Fisher curvature will be approximately the same as the loss curvature?
 - I am unclear about the strength of the intriguing link between low curvature directions and memorisation, what is the theoretical justification behind this? How do we know these directions not correspond to just noise?
 - Further, the explanation for why K-FAC specifically captures the memorisation/generalisation distinction better than other decompositions (e.g., SVD) could be stronger. The explanation about averaging canceling idiosyncratic directions is intuitive but lacks formal analysis.


I am happy to reconsider my score if my issues and questions can be addressed.

**Questions:**

- How are the energy threshold for eigenvector selection determined from the range of 60-90%? Are there observable gaps in the eigenvalues separating high and low curvature directions?
- Why is editing the specific layers selected the most effective, as are possible hypotheses for the observations from the sweeps across layers? Are there any memorisation specific eigenvectors that are consistent across layers or show some pattern?

---

> ### Author Response · Authors · 2025-12-04
>
> > How do we know that the Fisher curvature will be approximately the same as the loss curvature?
>
> Please see Appendix B in the updated draft (for now, we are keeping this in the appendix). For a well-trained model, the Fisher is a PSD approximation of the loss curvature. This is quite well understood in the relevant literature. For space we will leave it at that, but are happy to discuss this more. Our mild assumption is that the model is near a minimum, which we think is pretty fair.
>
>
> > Since the layer-wise K-FAC blocks ignore cross-layer effects, can you comment on how relevant such cross-layer correlations and coupling between attention weights and the weights of a layer's MLP would be? Is this a limitation of the proposed approach, or it is not relevant?
>
> This is a good point we don’t directly address, thank you for raising it. It is relevant, but we show that we can get clear signals without needing to accommodate cross-layer effects. It adds quite a bit of complexity to consider cross layer effects, so we believe it is beyond the scope of this work right now, but we will add this as a limitation and proposal for future work.
>
> A point about MLPs/Attention: MLP weights are about 66% of model parameters, replicable prior work connects MLP weights to memorization/fact retrieval. The only part of attention weights that may contain memorized information would be the Value and Output matrices. Given that these are such a small part of the overall parameter count, we consider it acceptable to forgo using them in our analysis for simplicity.
>
>
> > Why is editing the specific layers selected the most effective, as are possible hypotheses for the observations from the sweeps across layers?
>
> Yes, we might not make this point super explicitly, but the layers that the sweep picks are among those that have the greatest activation ratio difference between memorization and clean data. In particular, the sweep picks layers 23-25 in OLMo 2 7B. In Figure 2, we show a wide divergence there. In ViT, the sweep chooses layers 0 and 11. Again, Figure 2 shows very clearly this is where the widest divergences are
>
>
> > I am unclear about the strength of the intriguing link between low curvature directions and memorisation, what is the theoretical justification behind this? How do we know these directions not correspond to just noise?
>
> > The explanation about averaging canceling idiosyncratic directions is intuitive but lacks formal analysis.
>
> Thank you for raising these points. There are two questions to answer here:
> Do we know the low directions are not just noise?
> “The explanation about averaging canceling idiosyncratic directions is intuitive but lacks formal analysis.” Can we support this empirically?
>
> For the first one, yes we can support that this is not noise. First, evidence already in the paper: Figure 4 shows that tasks ‘more like memorization’ specifically have much greater activation with the lower 50% of eigenvectors.
>
> Further evidence we added: We designed a new experiment to get directly at this, since it is worth supporting. We show that the highest activating singular vectors of individual memorized sequences are orthogonal between each other, but also tend to live at the very bottom of the spectrum.
>
> Also, the theoretical link between curvature and memorization is very strong; the connection between curvature and generalization going back to Flat Minima (Hochreiter and Schmidhuber, 1997)

---

### Official Review · Reviewer_KuuX · 2025-10-31

**Soundness:** 2
**Presentation:** 2
**Contribution:** 3
**Rating:** 4
**Confidence:** 2

**Summary:**

This paper investigates how memorization and reasoning are represented within the weight space of LMs and ViTs by the curvature spectrum of the Fisher Information Matrix (K-FAC). The authors argue that high-curvature directions encode generalizable structure while low-curvature directions capture memorized or brittle behavior. They use this insight to design a curvature-based weight-editing method that suppresses memorized data more effectively than BalancedSubnet (BSN) while preserving general reasoning and validation accuracy.

The work is moderately novel, and the framing of reasoning vs. memorization in curvature space is conceptually elegant. However, it remains primarily descriptive and offers limited mechanistic insight for the claimed conceptual link between curvature and reasoning. Highly encourage the authors to deepen the causal analysis and broaden the evaluation during the discussion.

**Strengths:**

1. The paper raises an important question about how memorization and reasoning differ in weight-space geometry.
2. The use of curvature information (via K-FAC) for weight editing is technically sound and computationally efficient.
3. The idea of connecting curvature spectra with reasoning robustness is conceptually appealing.
4. The study is comprehensive, covering both vision and language domains, and connects curvature geometry to downstream behavioral changes.
5. The analysis of task brittleness (memorization → math → reasoning) is an inspiring conceptual framing.

**Weaknesses:**

1. The central claim is correlational: curvature and memorization are shown to co-vary. No causal analysis shows that curvature drives memorization.
2. The interpretation of eigenvectors as memorization vs. generalization directions is speculative. No visualization or qualitative evidence showing low-curvature directions actually encodes memorized samples.
3. Only one LM and one ViT are tested, and the memorization benchmarks are simplistic (mainly factual recall). Broader datasets or real-world leakage metrics (e.g., red-teaming) would strengthen the claim.
4. The paper alternates between “FIM curvature,” “loss curvature,” and “eigenbasis of weights” without always clarifying the mathematical distinctions.
5. The “reasoning continuum” is described qualitatively but not quantified; reasoning differences could reflect dataset or token-frequency effects rather than curvature structure.

**Questions:**

1. How robust are the results to the energy threshold (ρ) used for selecting eigenvectors—does performance vary smoothly or abruptly with ρ?
2. For the reasoning tasks, did you control for differences in perplexity or token entropy when interpreting drops in performance?
3. Is the curvature direction stable across checkpoints or random seeds, or do eigenbases drift substantially?
4. Can the method identify memorized content without labeled “memorized” examples?
5. Are curvature patterns stable across checkpoints or do they drift with training dynamics?

---

> ### Author Response · Authors · 2025-12-04
>
> We'd like to thank the reviewer for their review and questions and we are glad they found it important and interesting.
>
> >Can the method identify memorized content without labeled “memorized” examples?
>
> This is a good question and is an experiment that we believe we left on the table. We ran an additional experiment showing that using **completely unsupervised data** we explore editing single layers that: 1.) Maintain low perplexity, and 2.) have an outsized impact on reducing memorization. Please see the general comment
>
> > The central claim is correlational: curvature and memorization are shown to co-vary. No causal analysis shows that curvature drives memorization.
>
> We don’t agree with this as a criticism. Our experiments show that making edits based on curvature does indeed have strong effects on memorization. We base our work on multiple empirical and theoretical works spanning many years that establish this link. This alone is not a causal explanation but we think it supports our claims. We don’t rule out that there is an invisible variable that directly controls memorization, but this seems unlikely given current evidence.
>
> > The interpretation of eigenvectors as memorization vs. generalization directions is speculative. No visualization or qualitative evidence showing low-curvature directions actually encodes memorized samples.
>
> This is not true and our results are not speculative. Please see Figure 4. I would also point the reviewer to our related work section and Appendix B, which goes over the strong theoretical and empirical link between curvature and memorization.
>
> > Only one LM and one ViT are tested
>
> We tested a 7B and 1B LM (from the same family), and trained dozens of ViTs on ImageNet. Establishing this result across very distinct modalities does more work than testing additional LMs or additional ViTs. Reasonable people can disagree on what constitutes sufficient evidence for generality.
>
>
> > memorization benchmarks are simplistic (mainly factual recall). Broader datasets or real-world leakage metrics (e.g., red-teaming) would strengthen the claim.
>
> This is not true, the memorization benchmarks are actual memorized pretraining examples and memorized historical quotes. We *additionally* test factual recall (as well as math and logical reasoning tasks). We are not sure what the reviewer means by broader datasets or real world leakage.
>
>
> > For the reasoning tasks, did you control for differences in perplexity or token entropy when interpreting drops in performance?
>
> Thank you for raising this. We use the same edited OLMo 2 7B as was used in Section 5, so it does indeed have good perplexity.
>
> > The paper alternates between “FIM curvature,” “loss curvature,” and “eigenbasis of weights” without always clarifying the mathematical distinctions.
>
> Thank you for raising this. We have clarified the language throughout.
>
> > The “reasoning continuum” is described qualitatively but not quantified;
>
> This is not true. We spend the *most* time quantifying these results. See figures 3, 4, and 5.

---

### Author Response · Authors · 2025-12-04
**General Comment**

We thank the reviewers for their comments. To be brief, we have made the following revisions:

* Added a completely unsupervised version of our method. This shows that our method can be used to suppress memorization without *any labeled data* unlike the baseline BSN. This provides additional differentiating value that reviewers felt was missing/could have been demonstrated with our method.


* Added a demonstration to better justify our explanation that flat directions represent memorization because they are *on average* very flat.

* Clarified the writing to present the method more simply, pushing the interesting and motivating theoretical/empirical justification for focusing on curvature to the appendix, for those that are most interested in understanding the “why”.

## Unsupervised version of the weight edit to suppress memorization
Without using data labeled as having been memorized, we can use signal from K-FAC eigenvectors to find effective layers to edit out memorization. Using unlabeled data, we find layers with the largest difference in activation strength between top and bottom eigenvectors (similar to Figure 2, though without labeled memorized data), and select that single layer for editing. We get decently smooth results: those with less separation remove less memorization. For the single edit at layer 30, we maintain low perplexity, and achieve 14% strict-memorization.

| layer_proj | score  | ppl_post | mem_strict |
|-----------|--------|----------|------------|
| L31.down  | 4.4761 | 28.64    | 0.108      |
| L30.down  | 3.7384 | 22.28    | 0.14       |
| L31.up    | 3.3671 | 25.0     | 0.126      |
| L29.down  | 3.3205 | 20.63    | 0.254      |
| L14.up    | 3.2183 | 19.5     | 0.614      |
| L0.up     | 3.2106 | 20.8     | 0.578      |
| L13.up    | 3.0906 | 19.5     | 0.724      |
| L12.up    | 3.0616 | 19.7     | 0.706      |
| L11.up    | 3.0418 | 19.8     | 0.760      |
| L10.up    | 2.9962 | 20.0     | 0.694      |
| L15.up    | 2.9828 | 19.8     | 0.558      |
| L18.up    | 2.9171 | 20.5     | 0.302      |
| L8.up     | 2.8838 | 20.2     | 0.704      |
| L0.down   | 2.8751 | 19.37    | 0.69       |



## Are individual memorization directions “averaged out” to become flat?
We received a few comments about whether we could show experimentally that our claim that individually sharp directions that represent memorization get averaged out into flat directions, explaining our choice to edit these weight directions.

As context: prior work shows that for a single example, the loss curvature will be very sharp around that datum. In our work, we target memorized data by editing the directions that are on average *flat* in weight space. We claim that this is because the individually sharp directions are mostly orthogonal to each other, and on average have flat curvature across the whole dataset. Moderately curved directions across many data thus appear as the ‘sharpest’ on average.

Given a few random examples of memorized sequences in OLMo 2 7B, we can find activation-side eigenvectors that most strongly activate with hidden states at the last token position

Specifically, we take the activation-side K-FAC eigenvector v_j (column j of eigenvec_A), and compute alpha = |v_j^T x|. Results are below:

| j=10893 | j=11004 | j=10961 | j=10997 | j=10987 |
|---------|---------|---------|---------|---------|
| 12.0185 | 0.6512  | 1.2711  | 2.0682  | 0.2725  |
| 0.1183  | 10.2656 | 0.6369  | 4.1645  | 0.0477  |
| 0.3904  | 3.6096  | 8.0176  | 1.6783  | 0.7477  |
| 0.6089  | 0.6725  | 0.7659  | 10.7102 | 1.1138  |
| 0.3441  | 0.0489  | 2.7762  | 1.5473  | 7.7589  |

We show 5 individual memorized examples with high activation along a single direction j. The high diagonal compared to off-diagonal values shows that for each memorized example, their highest activating weight directions are unrelated to the highest activating direction for other memorized examples. Thus, these directions are idiosyncratic in representing one (or perhaps a few) examples strongly, but do not generally interact with other memorized examples much at all. I.e., on *average*, they are not generally activated.

---

### Meta-Review · Area_Chair_VJGE · 2025-12-13

**Summary:**

The paper provides an interesting perspective on connecting the curvature spectrum of the Fisher Information Matrix (K-FAC) with memorization. The reviewers acknowledge that this is an interesting, sound, and effective approach. They also see high potential of this work to inspire for future research.

At the same time, there are concerns about the limited evaluation setup (only two models tested), the lack of additional baselines, and cross-layer effects.

**Reviewer Concerns:**

During the rebuttal, the authors were able to address the most pressing concern, namely the reliance on labeled memorized data. The rebuttal introduced additional experiments showing that the method is effective even without such labeled data, which significantly strengthens the claims.

Overall, I see the following issues still outstanding (including additions and reading suggestions added by the AC after carefully reviewing the paper):

1. **Limited model choice**: Overall, the paper evaluates on a generative (Olmo) language model and a ViT vision classifier. While it is interesting to include both modalities, the generalization to different model families, model sizes, and architectures in the respective domains are absent. Furthermore, it is not well justified why for the language domain, a generative model was chosen, whereas for the vision domain, "only" a classifier was evaluated.
2. **Cross-layer effects not quantified:** Indeed, as mentioned by Reviewer Qin4, there is no quantification on how cross-layer effects, or accumulative effects. In fact, prior work [1] has shown that memorization accumulates between layers and that looking at per-layer memorization (without taking other layers into account) can lead to misleading claims about memorization. Therefore, at least considering the effects and carefully discussing them shall quantification not be possible, would be desirable.
3. **Requirement of a sweep**: There was a concern about the necessity of the sweep, which I think the authors can easily wave off for a future submission by benchmarking the timing and showing little overhead induced by this practice.
4. **Choice of baseline**: The paper solely compares against BSN, claiming that it is the strongest baseline. However, this does not acknowledge methods like [2]. Additionally, reading the described methodology (lines 204 and onward): "The way we measure this is through activation ratios: for some hidden activation xmem stemming from a memorized input, we compare the ratio of its activation with a weight component c, which is some direction in the input space of the weight matrix to the activation with a clean input xclean", this seems to be nearly exactly what is done in [3]. Discussing the differences and comparing as much as possible against [2,3] will strengthen the submission.
5. **Metrics used to evaluate ViT memorization (Concern by the AC)**: The 3 metrics used to quantify memorization in the classifier do not align with the state-of-the-art [4] compares against reference models (rather than measuring the direct drop in prediction confidence). This allows to factor the sample difficulty into account. Using (computationally tractable) derivatives of that leave-one-out style of memorization seem a more adequate choice for measuring memorization.




** References**

[1] Wang, Wenhao, Adam Dziedzic, Michael Backes, and Franziska Boenisch. "Localizing memorization in ssl vision encoders." Advances in Neural Information Processing Systems 37 (2024): 60475-60516.

[2] Chavhan, Ruchika, Ondrej Bohdal, Yongshuo Zong, Da Li, and Timothy Hospedales. "Memorization is Localized within a Small Subspace in Diffusion Models." In International Conference on Machine Learning (ICML)-Workshop on Generative AI and Law. 2024.

[3] Hintersdorf, Dominik, Lukas Struppek, Kristian Kersting, Adam Dziedzic, and Franziska Boenisch. "Finding nemo: Localizing neurons responsible for memorization in diffusion models." Advances in Neural Information Processing Systems 37 (2024): 88236-88278.

[4] Feldman, Vitaly. "Does learning require memorization? a short tale about a long tail." In Proceedings of the 52nd annual ACM SIGACT symposium on theory of computing, pp. 954-959. 2020.

**Reviewer Scores:**

I expect the reviewer scores to have stayed roughly the same.

---

### Decision · Program_Chairs · 2026-01-26

Reject